

**Air Quality and Climate Change, Topic 3 of the Model Inter-Comparison**
**Study for Asia Phase III (MICS-Asia III), Part I: overview and model**
**evaluation**
Meng Gao[1,2], Zhiwei Han[3,4], Zirui Liu[5], Meng Li[6, 13], Jinyuan Xin[5], Zhining Tao[7,8], Jiawei Li[4], Jeong-Eon
Kang[9], Kan Huang[10], Xinyi Dong[10], Bingliang Zhuang[11], Shu Li[11], Baozhu Ge[5], Qizhong Wu[12], Yafang
Cheng[13], Yuesi Wang[5], Hyo-Jung Lee[9], Cheol-Hee Kim[9], Joshua S. Fu[10], Tijian Wang[11], Mian Chin[8],
Jung-Hun Woo[14], Qiang Zhang[6], Zifa Wang[4,5], Gregory R. Carmichael[1]
1 Center for Global and Regional Environmental Research, University of Iowa, Iowa City, IA, USA
2 John A. Paulson School of Engineering and Applied Sciences, Harvard University, Cambridge, MA, USA
3 Key Laboratory of Regional Climate-Environment for Temperate East Asia, Institute of Atmospheric Physics,
Chinese Academy of Sciences, Beijing, China
4 University of Chinese Academy of Sciences, Beijing 100049, China
5 State Key Laboratory of Atmospheric Boundary Layer Physics and Atmospheric Chemistry, Institute of
Atmospheric Physics, Chinese Academy of Sciences, Beijing, China
6 Ministry of Education Key Laboratory for Earth System Modeling, Center for Earth System Science, Tsinghua
University, Beijing, China
7 Universities Space Research Association, Columbia, MD, USA
8 NASA Goddard Space Flight Center, Greenbelt, MD, USA
9 Department of Atmospheric Sciences, Pusan National University, Busan, South Korea
10 Department of Civil and Environmental Engineering, University of Tennessee, Knoxville, TN, USA
11 School of Atmospheric Sciences, Nanjing University, Nanjing, China





12 College of Global Change and Earth System Science, Beijing Normal University, Beijing, China
13 Multiphase Chemistry Department, Max Planck Institute for Chemistry, Mainz, Germany
14 Department of Advanced Technology Fusion, Konkuk University, Seoul, South Korea
Correspondence to: M. Gao (mgao2@seas.harvard.edu), Z. Han (hzw@mail.iap.ac.cn), and G. R.
Carmichael (gcarmich@engineering.uiowa.edu)

## Abstract

Topic 3 of the Model Inter-Comparison Study for Asia (MICS-Asia) Phase III examines how
online coupled air quality models perform in simulating high aerosol pollution in the North
China Plain region during wintertime haze events and evaluates the importance of aerosol
radiative and microphysical feedbacks. A comprehensive overview of the MICS-ASIA III Topic
3 study design, including descriptions of participating models and model inputs, the experimental
designs, and results of model evaluation, are presented. Two winter months (January 2010 and
January 2013) were selected as study periods, when severe haze occurred in North China.
Simulations were designed to evaluate radiative and microphysical feedbacks, together and
separately, relative to simulations without feedbacks. Six modeling groups from China, Korea
and the United States submitted results from seven applications of online coupled chemistry-
meteorology models. Results are compared to meteorology and air quality measurements,
including the Campaign on Atmospheric Aerosol Research Network of China (CARE-China)
network, and the Acid Deposition Monitoring Network in East Asia (EANET). The analysis
focuses on model evaluations and aerosol effects on meteorology and air quality, and potentially
other interesting topics, such as the impacts of model resolutions on aerosol-radiation-weather



interactions. The model evaluations for January 2010 show that current online-coupled
meteorology-chemistry model can generally well reproduced meteorological features and
variations of major air pollutants, including aerosol concentrations. The correlation coefficients
between multi-model ensemble mean and observed near-surface temperature, water vapor
mixing ratio and wind speeds can reach as high as 0.99, 0.99 and 0.98. The correlation
coefficients between multi-model ensemble mean and the CARE-China observed near-surface
air pollutants range from 0.51 to 0.94 (0.51 for ozone and 0.94 for $PM_{2.5}$). However, large
discrepancies exist between simulated aerosol chemical compositions from different models,
which is due to different parameterizations of chemical reactions. The coefficient of variation
(standard deviation divided by average) can reach above 1.3 for sulfate in Beijing, and above 1.6
for nitrate and organic aerosol in coastal regions, indicating these compositions are less
consistent from different models. During clean periods, simulated Aerosol Optical Depths
(AOD) from different models are consistent, but peak values differ during severe haze event,
which can be explained by the differences in simulated inorganic aerosol concentrations and the
hygroscopic growth efficiency (affected by varied RH). These results provide some brief senses
of how current online-coupled meteorology-chemistry models reproduce severe haze events, and
some directions for future model improvements.

## 1 Introduction

Air pollution in Asia, particularly in China and India, has been an increasing important research
topic, and has attracted enormous media coverage since about 60% of the world population live
and are exposed to extremely unhealthy air in this region. It is estimated that outdoor air





pollution brings about 3.3 million premature deaths per year worldwide but primarily in Asia
(Lelieveld et al., 2015). In addition, the impacts of regional and intercontinental transport of
Asian pollutants on air quality and climate change have been frequently reported (Akimoto,
2003; Menon et al., 2002, Ramanathan and Carmichael, 2008). Chemical transport models have
been developed and applied to study various air pollution issues in Asia. For example, an
Eulerian regional scale acid deposition and photochemical oxidant model was developed in the
United States (Carmichael and Peters, 1984; Carmichael et al., 1986; Carmichael et al., 1991)
and applied to study long-range transport of sulfur oxides ($SO_x$), dust and ozone production in
East Asia (Carmichael et al., 1998; Xiao et al., 1997); a nested urban and regional scale air
quality prediction modeling system was developed and applied to investigate ozone pollution in
Taiwan (Wang et al., 2001). Although important advances have taken place in air quality
modeling, large uncertainties still remain, which are related to inaccurate and/or incomplete
emission inventories, poorly represented initial and boundary conditions and missing or poorly
parameterized physical and chemical processes (Carmichael et al., 2008a).
Furthermore, many models used to study air quality in Asia have been developed in other regions
(e.g., USA and Europe), and the assumptions and parameterizations included in these models
may not be applicable to the Asian environment. In order to develop a common understanding of
model performance and uncertainties in Asia, and to further develop the models for Asian
applications, a model inter-comparison study was initiated, i.e., Model Inter-Comparison Study
for Asia Phase I (MICS-Asia I), in 1998 during a workshop on Transport of Air Pollutants in
Asia in Austria. The focus of MICS-Asia Phase I was to study long-range transport and
deposition of sulfur within Asia in support of on-going acid deposition studies. Eight long-range
transport models from six institutes in Korea, Japan, Denmark, the USA, and Sweden



participated in MICS-Asia I. Multi-model results of sulfur dioxide ($SO_2$) and sulfate
concentrations, and wet deposition amounts in January and May 1993 were compared with
surface observations in East Asia (Carmichael et al., 2002). Source-receptor relationships and
how model structure and parameters affect model performance were also discussed during this
phase (Carmichael et al., 2002). In 2003, MICS-Asia Phase II was initiated to include more
species, including nitrogen compounds, ozone and aerosols. The study period was expanded to
cover two different years and three different seasons, and global inflow to the study domain was
also considered (Carmichael et al., 2008b). Nine modeling groups from Korea, Hong Kong,
Japan, the USA, Sweden, and France participated in this phase. Seven topics (i.e., ozone and
related precursors, aerosols, acid deposition, global inflow of pollutants and precursors to Asia,
model sensitivities to aerosol parameterization, analysis of emission fields, and detailed analyses
of individual models) were discussed and published in a special issue of Atmospheric
Environment (Carmichael et al., 2008b).
In 2010, MICS-Asia phase III was launched and three topics for this phase were decided during
the first and second Workshop on Atmospheric Modeling in East Asia. Phase III aims to evaluate
strengths and weaknesses of current air quality models and provide techniques to reduce
uncertainty in Asia (Topic 1), to develop a reliable anthropogenic emission inventory in Asia
(Topic 2), and to evaluate aerosol-weather-climate interactions (Topic 3). Various multi-scale
models participated in this phase and the study periods range from year to month depending on
study topics. This phase uses data from the Acid Deposition Monitoring Network in East Asia
(EANET), in addition to new observations related to atmospheric chemistry in the region. An
important advance to this phase is the inclusion of multiple online-coupled chemistry-
meteorology models to investigate aerosol-weather-climate interactions, which is the target of





topic 3. On-line coupled models are playing important roles in air quality, meteorology and
climate applications, but many important research questions remain (Baklanov et al., 2017).
The influences of aerosols on meteorology, e.g., radiation, temperature, boundary layer heights,
winds, etc. and PM$_{2.5}$ concentrations have been examined in previous studies using different
online coupled models (Gao et al., 2016a, 2016b; Han et al., 2012; Tao et al., 2015, 2016; Wang
et al., 2014; Zhang et al., 2010). In general, there are two ways of online coupling: online
integrated coupling (meteorology and chemistry are simulated using the same model grid, and
one main time step is used to integrate) and online access coupling (meteorology and chemistry
are independent but data are exchanged on a regular basis) (Baklanov et al., 2014). These two
different coupling ways can lead to uncertainties in the results of aerosol-weather-climate
interactions. Even using the same coupling way, different parameterizations in different online
models causes uncertainties as well. Thus, it is important to inter-compare how different online
models simulate aerosol-weather-climate interactions.
This paper presents and overview of the MICS-ASIA III Topic 3, serving as the main repository
of the information linked to Topic 3 simulations and comparisons. This paper is organized as
follows: in Section 2, we provide the inter-comparison framework of Topic 3, including the
participating models, emissions, boundary conditions, observational data, and analysis
methodology. Section 3 presents the general descriptions of the study periods and Section 4
presents comparisons and discussions focused on the results related to the meteorological and air
pollution conditions during the January 2010 heavy haze episode. The results of January 2013
haze episode and detailed analysis of the direct and indirect effects will be presented in a
companion paper.



## 2 Inter-comparison framework

In North China, severe aerosol pollution frequently happens and attracts enormous interests from

both public and scientific communities (Cheng et al., 2016; Gao et al., 2015, 2016a, 2016b,

2016c, 2017). Two winter months in which severe haze episodes happened in North China were

selected as the study periods for Topic 3. During these two months, maximum hourly $PM_{2.5}$

concentration in urban Beijing reached ~500 µg/m$^3$ and 1000µg/m$^3$, respectively. Compared to

the China Grade 1 24-h $PM_{2.5}$ standard (35µg/m$^3$), daily mean $PM_{2.5}$ concentrations in urban

Beijing exceeded this standard for 20 days and 27 days within these two months, respectively.

The dramatically high aerosol loadings during these two hazy months substantially affected

radiation transfer, and provide a good opportunity to study the aerosol effects on weather, air

quality and climate. In this study, the participants were required to use common emissions to

predict air quality during these two months and submit requested model variables. The emissions

were placed on a publicly accessible website. Six modeling groups submitted results for Topic 3.

In this section, we briefly describe these models and their configurations, introduce the emission

inventories (including anthropogenic, biogenic, biomass burning, air and ship, and volcano

emissions), observational datasets, and describe the analysis methodology.

### 2.1 Participating models

Table 1 summarizes the characteristics of the participating models. These models include: one

application of the Weather Research Forecasting model coupled with Chemistry (WRF-Chem,

Grell et al., 2005) by Pusan National University (PNU) (M1), one application of the WRF-Chem

model by the University of Iowa (UIOWA) (M2), two applications (two domains: 45km and

15km horizontal resolutions) of the National Aeronautics and Space Administration (NASA)



Unified WRF (NU-WRF, Peters-Lidard et al., 2015; Tao et al., 2013) model by the Universities
Space Research Association (USRA) and NASA's Goddard Space Flight Center (M3 and M4),
one application of the Regional Integrated Environment Modeling System with Chemistry
(RIEMS-Chem, Han et al., 2010) by the Institute of Atmospheric Physics (IAP), Chinese
Academy of Sciences (M5), one application of the coupled Regional Climate Chemistry
Modeling System (RegCCMS, Wang et al., 2010) from Nanjing University (M6), and one
application of the coupled WRF-CMAQ (Community Multiscale Air Quality) model by the
University of Tennessee at Knoxville (UTK) (M7). These models are all online coupled, which
enables aerosol-weather-climate interactions. Domain setting of each model application is shown
in Figure 1. The domains of M2, M5, and M6 (UIOWA, IAP, and NJU in Figure 1) cover most
areas of East Asia, including China, North Korea, South Korea, Japan, Mongolia, and north parts
of Southeast Asia. M1, M3 and M7 domains (PNU, NASA D01 and UTK) include more
countries in Southeast and South Asia. M4 (NASA D02) covers east China, Korea and Japan.
The horizontal model resolutions of these applications range from 15km to 60km (Table 1).
Model vertical resolutions vary from 16 to 60 layers (Table 1) and the set model top pressures
range from 100mb to 20mb.
Gas phase chemistry and aerosol modules are key components of chemical transport models.
Although the WRF-Chem and NU-WRF models were applied at three institutes (PNU, UIOWA,
and NASA), different gas phase chemistry and aerosol modules were used. At PNU (M1), the
RACM-ESRL (Regional Atmospheric Chemistry Mechanism, Earth System Research
Laboratory) gas phase chemistry coupled with MADE/VBS (Modal Aerosol Dynamics Model
for Europe/Volatility Basis set) aerosol module was used. RACM was developed based on
Regional Acid Deposition Model (RADM2) to simulate regional atmospheric chemistry



(Stockwell et al., 1997) (including 237 reactions) and the rate coefficients were updated in
RACM ESRL version (Kim et al., 2009). MADE uses 3 log-normal modes (Aitken,
accumulation, coarse) and simulates major aerosol compositions, including sulfate, ammonium,
nitrate, sea-salt, black carbon (BC), and organic carbon (OC). In addition, the VBS method was
implemented to simulate secondary organic aerosols (SOA). At the University of Iowa (M2),
CBMZ (Carbon-Bond Mechanism version Z) gas phase chemistry coupled with an 8 bin
MOSAIC (Model for Simulating Aerosol Interactions and Chemistry) aerosol module was
applied. CBMZ (Zaveri and Peters, 1999) extends the original CBM4 mechanism to function
properly at larger spatial and longer timescales. The augmented CBMZ scheme includes 67
species and 164 reactions. MOSAIC considers major aerosol species at urban, regional and
global scales, including sulfate, nitrate, ammonium, sodium, chloride, EC, and other unspecified
inorganic species (such as inert minerals, trace metals, and silica) (Zaveri et al. 2008). MOSAIC
includes some aqueous reactions but no SOA formation. At NASA, the GOCART aerosol model
(Chin et al., 2002) was coupled to RADM2 gas phase chemistry, and incorporated into the NU-
WRF model (M3 and M4) to simulate major tropospheric aerosol species, including sulfate, BC,
OC, dust, and sea-salt. In this aerosol model, 10% of organic compounds from the volatile
organic compounds (VOCs) emission inventory are assumed to be converted to SOA (Chin et al.,

196    2002).

Both the RIEMS-Chem model from IAP (M5) and the RegCCMS model from NJU (M6) used
CBM4 to calculate gas phase chemistry (Gery et al., 1989). The CBM4 version incorporated in
RIEMS-Chem (M5) includes 37 species and 91 reactions, and aerosols in RIEMS-Chem include
sulfate, nitrate, ammonium, BC, OC, SOA, 5 bins of soil dust, and 5 bins of sea salt (Han et al.,
2012). ISORROPIA (Nenes et al., 1998) is coupled to RIEMS-Chem to treat thermodynamic





equilibrium process and to simulate inorganic aerosols. SOA production from primary
anthropogenic and biogenic VOCs is calculated using a bulk aerosol yield method according to
Lack et al. (2004). A lognormal size distribution is assumed for inorganic aerosols, BC, and OC,
with median radius of 0.07 mm, 0.01 mm, and 0.02 mm, and geometric standard deviation of 2.0,
2.0, and 2.2, respectively. The schemes for soil dust deflation and sea salt generation were from
Han et al. (2004), which used 5 size bins (0.1-1.0, 1.0-2.0, 2.0-4.0, 4.0-8.0, 8.0-20.0μm) to
represent dust and sea salt size distribution. The refractive indices of aerosol components were
mainly derived from the OPAC (Optical Properties of Aerosols and Clouds) database. Aerosol
extinction coefficient as well as single scattering albedo and asymmetry factor are calculated by a
Mie-theory based parameterization developed by Ghan and Zaveri (2007), which has a high
computational efficiency with similar degree of accuracy compared with complete Mie code. An
internal mixture of aerosols was assumed in this region of large emissions. A method known as
kappa (k) parameterization (Petters and Kreidenweis, 2007) was adopted to represent the aerosol
hygroscopic growth.
The version of CBM4 implemented in RegCCMS (M6) consists of 36 reactions (4 photolysis
reactions) and 20 species (Wang et al., 2010). RegCCMS also used ISORROPIA to calculate
inorganic aerosols (Wang et al., 2010). For implementation of aerosol effects, sulfate radiative
properties were treated following Kiehl and Briegleb (1993), OC were assumed to have the same
properties as sulfate, and the wavelength-dependent radiative properties of BC follows Jacobson

221    (2001).

M7 applied SAPRC 99 coupled to the sixth-generation CMAQ aerosol module (AE6) to simulate
gas phase chemistry and aerosol formation The SAPRC99 mechanism implanted within the
CMAQ model has 88 species and 213 chemical reactions (Carter, 2000a,b). AE6 aerosol





mechanism is to couple with WRF. There are seven components including water soluble mass,
water insoluble mass, elemental carbon, sea salt, water, diameters and standard deviations passed
to WRF. Many previous studies have underscored that the choice of gas phase mechanism and
aerosol models are of great importance for simulating air pollutants (Knote et al., 2015; Zhang et
al., 2012).  The different gas phase chemistry and aerosol modules used in the participating
models are expected to yield notable differences in performances, which are shown later in
section 4.
Although the WRF-Chem and NU-WRF models were applied at three institutes (PNU, UIOWA,
and NASA), different physics configurations are also used. Table S1 compares the used
microphysics, radiation, boundary layer, cumulus clouds, and surface schemes used by WRF-
Chem and NU-WRF applications. Both aerosol-radiation and aerosol-microphysics are included
in these applications, but different used microphysics (M1 uses Lin, M2 uses Morrison and M3
uses Goddard) and radiation (M1 and M2 use RRTMG, but M3 uses Goddard) schemes will lead
to differences in estimates in aerosol direct and indirect effects, in addition to the differences in
simulated aerosols.
**2.2 Emissions**
The accuracy of air quality modeling results highly depends on the quality and reliability of
emission inventory. Accordingly, a new Asian emission inventory was developed for MICS-III
by integrating state-of-the-art national/regional inventories to support this model inter-
comparison study (Li et al., 2017). This is the major theme of MICS-ASIA III Topic 2. These
emissions, along with biogenic emissions, biomass burning emissions, emissions from air and



ship, and volcano emissions were used. This section offers some basic descriptions of these
provided emissions.

**2.2.1 Anthropogenic emissions**

The state-of-the-art anthropogenic emission inventory for Asia (MIX) was developed by
incorporating five inventories, including the REAS inventory for Asia developed at the Japan
National Institute for Environmental Studies (NIES), the MEIC inventory for China developed at
Tsinghua University, the high resolution ammonia ($NH_3$) emission inventory in China developed
at Peking University, the Indian emission inventory developed at Argonne National Laboratory
in the United States, and the CAPSS Korean emission inventory developed at Konkuk University
(Li et al., 2017). This MIX inventory includes emissions for ten species, namely $SO_2$, nitrogen
oxides ($NO_x$), carbon monoxide (CO), non-methane volatile organic compounds (NMVOC),
$NH_3$, $PM_{10}$, $PM_{2.5}$, BC, OC, and carbon dioxide ($CO_2$). NMVOC are provided with CB-05 and
SAPRC-99 speciation datasets. Emissions of these species were prepared for years 2008 and
2010 in monthly temporal resolution and 0.25 degree spatial resolution. Weekly/diurnal profiles
were also provided. Five sectors were considered, namely industry, power generation, residential
sources, transportation and agriculture. Figure 2 shows the spatial maps of these ten species for
January 2010. Emissions of most of these species exhibit similar spatial patterns, with enhanced
values in east China and lower values in north and south India. Emissions of $NH_3$ display a
different spatial distribution, with pronounced values in India and lower values in north China
(Figure 2). More detailed description of this emission inventory is documented in Li et al.

266    (2017).

**2.2.2 Biogenic emissions**



Terrestrial ecosystems generate miscellaneous chemical species, including volatile and semi-
volatile compounds, which play important roles in atmospheric chemistry and are the largest
contributor to global annual flux of reactive volatile organic compounds (VOCs) (Guenther et
al., 2006). For MICS-ASIA III, hourly biogenic emissions were provided for the entire year of
2010 using the Model of Emissions of Gases and Aerosols from Nature (MEGAN) version 2.04
(Guenther et al., 2006). The variables that drive MEGAN include land cover information (plant
function type, leaf area index) and weather condition, which includes solar transmission, air
temperature, humidity, wind speed, and soil moisture. In the preparation of MEGAN biogenic
emissions, land cover information is taken from the NASA MODIS products, and weather
condition are calculated using WRF simulations. Figure S1 shows biogenic emissions of some
selected species (isoprene and HCHO) for January 2010. High biogenic emissions are found in
south Asia during winter, including India, south China, and Southeast Asia, where solar
radiation, air temperature and vegetation covers are relatively higher than in northern regions.
Some models used these emissions directly. Others internally calculated the biogenic emissions
on-line with the model predicted meteorology using the MEGAN model.
**2.2.3 Biomass burning emissions**
Biomass burning in the tropics is a strong contributor to air pollutants, and extensive biomass
burning in Asia, particularly Southeast Asia, exerts a great influence on air quality (Streets et al.,
2003). For MICS-ASIA III, biomass burning emissions were processed by re-gridding the Global
Fire Emissions Database version 3 (GFEDv3) (0.5 by 0.5 degree). GFED fire emissions are
estimated through combining satellite-detected fire activity and vegetation productivity
information. Carbon, dry matter, $CO_2$, CO, $CH_4$, hydrogen, nitrous oxide, $NO_x$, NMHC, OC, BC,
$PM_{2.5}$, total particulate matter and $SO_2$ emissions are estimated in monthly temporal resolution.



Figure S2 shows the gridded biomass burning emissions for January 2010. Biomass burning
activity is highest in Cambodia and some areas of Myanmar and north of Thailand (Figure S2),
and the peak emission season is spring. Although it has been concluded that biomass burning
could significantly contribute to aerosol concentrations in China, the contribution is limited for
Topic 3 study since the focused region is North China where biomass burning emissions are
negligible during cold winter (Gao et al., 2016a).
**2.2.4 Volcanic SO$_2$ emissions**
Volcanoes are important sources of various sulfur and halogen compounds, which play crucial
roles in tropospheric and stratospheric chemistry. It is estimated that SO$_2$ emitted from volcanoes
account for about 9% of the total worldwide annual SO$_2$ flux (Stoiber et al., 1987). The Asia-
Pacific region is one of the most geologically unstable regions in the world where many active
volcanoes are located. During MICS-ASIA Phase II, the volcano SO$_2$ emissions had already
been provided for chemical transport models (Carmichael et al. 2008b). Volcano SO$_2$ emissions
were provided, with a daily temporal resolution. In January, some volcanoes in Japan are very
active, such as Miyakejima (139.53ºE, 34.08ºN, and 775m above sea level) and Sakurajima
(130.65ºE, 31.59ºN, 1117m above sea level).
**2.2.5 Air and Ship emissions**
Fuel burning in aircraft and ship engines produces greenhouse gases and air pollutants. The
shipping and aircraft emissions used are based on HTAPv2 emission inventory (0.1 by 0.1
degree) for year 2010 (Janssens-Maenhout et al., 2015), provided on an annual basis. Aircraft
emissions include three parts: landing and takeoff (LTO), climbing and descent (CDS), and
cruise (CRS). Aircraft emission hot spots are mostly located in Japan, and Beijing, Yangtze



River Delta (YRD) and Pearl River Delta (PRD) in China (Figure S3). East China Sea, sea
around Japan and Singapore exhibit high shipping emissions due to active shipping
transportation (Figure S3). It is estimated that international shipping contributed about 10% to
the global $SO_2$ emissions, and together with aviation contribute more than 10% of global $NO_x$
emissions (Janssens-Maenhout et al., 2015).
**2.3 Boundary conditions**
To predict more realistic spatial and temporal variations of air pollutants, boundary conditions
from global chemical transport models are necessary to drive regional chemical transport models
(Carmichael et al., 2008b). Simulations of three global chemical transport models (i.g.,
CHASER, GEOS-Chem and MOZART) were provided as boundary conditions for MICS-ASIA
III. CHASER was developed in Japan to simulate the $O_3$-$HO_x$-$NO_x$-$CH_4$-CO photochemical
system and its effects on climate (Sudo et al., 2002). GEOS-Chem was developed in the USA to
simulate tropospheric chemistry driven by assimilated meteorology (Bey et al., 2001). In
addition, the National Center for Atmospheric Research (NCAR) also provides global
simulations of atmospheric chemistry (MOZART model) and an interface to convert them to
WRF-Chem boundary conditions (Emmons et al., 2010), and NASA provides global aerosol
distributions using the global GOCART chemistry model (Chin et al., 2002). GEOS-Chem was
run with 2.5°x2° resolution and 47 vertical layers and CHASER model was run with 2.8°x2.8°
and 32 vertical layers. 3 hourly-average fields of gaseous and aerosols were distributed to all
participants. The MOZART-4 simulations were also configured at the horizontal resolution of
2.8°x2.8°, but with 28 vertical levels. NASA GOCART was configured at the same resolution as
GEOS-5 meteorology (1.25°x1°). As listed in Table 1, M1 used climatological data from the
NOAA Aeronomy Lab Regional Oxidant Model (NALROM), while M2 used boundary



conditions from the MOZART-4 (provided from the NCAR website). M3 and M4 used
MOZART-4 as boundary conditions for gases and used GOCART as boundary conditions for
aerosols. M6 also used fixed climatology boundary conditions, and M5 and M7 used GEOS-
Chem outputs as boundary conditions. Even though the same global model is used as boundary
conditions, the treatments of inputs might differ in details, which might lead to considerable
dissimilarities. In MICS-ASIA II, Holloway et al. (2008) discussed the impacts of uncertainties
in global models on regional air quality simulations.
**2.4 Observation data**
Historically, the lack of reliable air quality measurements in Asia has been a bottleneck in
understanding air quality and constraining air quality modeling in Asia. Beginning MICS-ASIA
II, observational data from Acid Deposition Monitoring Network in East Asia (EANET) has
been used to evaluate model performance. EANET was launched in 1998 to address acid
deposition problems in East Asia, following the model of the Cooperative Program for
Monitoring and Evaluation of the Long-range Transmission of Air pollutants in Europe (EMEP).
As of 2010, there are 54 wet deposition sites and 46 dry deposition sites in 13 participating
countries. Quality assurance and quality control measures are implemented at the national levels
and in the Inter-laboratory Comparison Project schemes to guarantee high quality dataset.
EANET supported current activities of MICS-ASIA III, and provided measurements in 2010 to
all modeling groups. More information about EANET dataset can be found in
http://www.eanet.asia/.
In addition to EANET data, measurements of air pollutants and aerosol optical depth (AOD)
collected at the Campaign on Atmospheric Aerosol Research network of China (CARE-China)



(Xin et al., 2015) network were also used. Previous successful networks in Europe and the
United States underscored the importance of building comprehensive observational networks of
aerosols in China to get better understanding of the physical, chemical and optical properties of
atmospheric aerosols across China. As the first comprehensive attempt in China, CARE-China
was launched in 2011 by Chinese Academy of Sciences (CAS) (Xin et al., 2015). Before
launching this campaign, CAS had already been measuring air pollutants and AOD at some
CARE-China sites. Table 2 summaries the locations and characteristics of the CARE-China
measurements for January 2010. Air quality measurements include concentrations of $PM_{2.5}$,
$PM_{10}$, $SO_2$, $NO_2$, NO, CO, $O_3$.
In addition, AOD from Aerosol Robotic Network (AERONET) (https://aeronet.gsfc.nasa.gov/)
and operational meteorological measurements (near surface temperature, humidity, wind speed
and downward shortwave radiation) in China and atmospheric sounding data in Beijing were
used. AERONET provides long-term, continuous, readily accessible and globally distributed
database of spectral AOD, inversion products and precipitable water. AOD data are calculated
for three quality levels: Level 1.0 (unscreened), Level 1.5 (cloud screened), and Level 2.0 (cloud
screened and quality assured) (Holben et al., 1998). The locations and characteristics of the
AERONET measurements are also summarized in Table 2. In-situ measurements of
meteorological data from standard stations in China are operated by China Meteorological
Administration (CMA) and different levels of data, including daily, monthly, and annually, are
open to the public (http://data.cma.cn/en). The locations of all used observational sites are
marked in Figure S4, Figure S5 and Figure S6.
The meteorology measurements (locations are shown in Figure S4) were averaged and compared
with model results that averaged across those locations. The radiation measurements were



averaged and compared against model results in North China and South China (locations are
shown in Figure S5), separately. The CARE-China, AERONET and EANET measurements
(locations are shown in Figure S5 and S6) were compared against model results site by site, and
model ensemble mean values were made by averaging all model results.
**2.5 Analysis methodology**
All groups participating in Topic 3 were requested to simulate meteorology, air quality, radiative
forcing and effects of aerosols over the Beijing-Tianjin-Hebei region of east China during two
periods: January 2010 and January 2013. Simulations were designed to evaluate radiative and
microphysical feedbacks, together and separately, relative to simulations without feedbacks.
Each group was requested to submit the following fields from their simulations.
(1) hourly mean meteorology:
(a) air temperature and water vapor mixing ratio at 2m above ground (T2, Q2), wind speed at
10m above groud (WS10), and shortwave radiation flux (Wm-2) at the surface;
(b) above variables (except shortwave radiation flux) at 1km and 3km above ground.
(2) hourly mean concentrations:
(a) $SO_2$, $NO_x$, CO, O3, $PM_{2.5}$, $PM_{10}$ and sulfate, nitrate, ammonium, BC, OC and dust in $PM_{2.5}$;
(b) above variables at 1km and 3km above ground.
(3) hourly mean AOD, aerosol direct radiative forcings at the surface, top of the atmosphere
(TOA) and inside the atmosphere (single scattering albedo is an option for participants).
(4) Hourly mean integrated liquid water, cloud optical depth.



(5) Changes in T2, Q2, WS10 and PM$_{2.5}$ concentrations at the surface due to both direct and
indirect aerosol's effects.
We calculated multiple model evaluation metrics, including correlation coefficient (r), root mean
square error (RMSE), mean bias error (MBE), normalized mean bias (NMB), mean fractional
bias (MFB) and mean fractional error (MFE). The equations are presented in supplemental
information.

**3 General description of meteorology and haze during the study periods**
Winter haze events are frequently happening in east China, which is partially due to the stagnant
weather conditions in winter. Here we present general descriptions of the meteorological
conditions during the selected two January months using the NCEP/NCAR reanalysis products.
Figure 3 (a, b) display the monthly mean T2 (temperature at 2m) and W10 (wind speeds at 10m)
for January 2010 and January 2013, respectively. For both periods, WS10 were very weak in
eastern and central China regions. T2 in Mongolia region was relatively higher for January 2013.
Historical analyses have shown that cold conditions are usually associated with strengthened
Siberian High (Gong and Ho, 2002), and relatively higher T2 and more weakened Siberian High
(Figure 3 (c, d)) during January 2013 led to weaker winter monsoon winds and higher pollution
levels. The relatively weaker Siberian High during January 2013 compared to January 2010 is
also shown in the sea level pressures (Figure 3 (c, d)). The Siberian High center was about
1037mb during January 2013, lower than that (1040mb) during January 2010. Figure 3 (c, d)
show that there was no significant precipitation in North China and heavy rainfall only occurred
in Southeast Asia regions. During cold winters, northern China burns coal for heating, generating





more emissions. Under stagnant weather conditions, haze episodes are easily triggered. It was
reported that January 2013 was the haziest month in the past 60 years in Beijing, and
instantaneous $PM_{2.5}$ concentration exceeded 1000µg/m$^3$ in some areas in Beijing. Winter haze
also happened from 16 to 19 January in 2010. High concentrations of aerosols during these two
study periods provide great opportunity to study aerosol-radiation-weather interactions.

**4 Results and discussions**
In this section, we present some major features of model performances in meteorological and
chemical variables for the January 2010 period. Detail analysis of feedbacks and radiative
forcing are presented in MICS-ASIA III companion papers. Heavy haze occurred over broad
regions of East China in January 2010. The plots of observed meteorological variables and $PM_{2.5}$
in Beijing show the general situation (Figure 4). Elevated $PM_{2.5}$ occurred during three periods
separated in time by roughly one week (January 8, 16 and 26). The major event occurred during
January 15-21. The events occurred during periods of low wind speeds, and increasing
temperature and relative humidity. The high $PM_{2.5}$ concentrations during January 15-21 also
greatly reduce the downward shortwave radiation.  Below we evaluate how well the models
predict these features.
**4.1 Evaluation of meteorological variables**
Air quality is affected by not only emissions, but also meteorological conditions. Meteorology
affects air quality through altering emissions, chemical reactions, transport and deposition
processes (Gao et al., 2016b). Thus, it is important to assess how well these participating models





reproduced meteorological variables. The predicted temperature at 2m high (T2), water vapor
mixing ratio at 2m (Q2), wind speed at 10m high (WS10) and daily maximum downward
shortwave radiation (SWDOWN) were evaluated against near surface observations at the CMA
sites.
Figure 5 (a-c) shows the comparisons between simulated and observed daily mean T2, Q2 and
WS10 averaged over stations in East China (locations are shown in Figure S4) during January
2010, along with multi-model ensemble mean and observation standard deviation. The calculated
correlation coefficients between models and observations are also shown in Figure 5 and other
calculated model evaluation metrics are summarized in Table 3. In general, the simulated
magnitudes and temporal variations of T2 and Q2 show high order of consistencies with
observations, with correlation coefficients ranging from 0.88 to 1. For T2, models tend to have a
cool bias; M1 and M2 have the lowest RMSE (0.64 and 0.68), lowest MBE (-0.19 and -0.60) and
lowest NMB (-0.07% and -0.22%) values (Table 3). For Q2, most models tend to slightly
overestimate; M1 and M2 have the best performance, with the lowest RMSE (0.14 and 0.10),
lowest MBE (0.02 and -0.01), and lowest NMB (0.84% and -0.55%) values (Table 3).
Simulated WS10 exhibit larger diversity of results. All models tend to overestimate WS10, with
MBE ranging from 0.15m/s to 2.37m/s. Overestimating wind speeds under low wind conditions
is a common problem of current weather forecasting models, and many factors, including errors
in terrain data and reanalysis data, relatively low horizontal and vertical model resolutions, as
well as poorly parameterized urban surface effect, contribute to these overestimations. From the
calculated RMSE, MBE, and NMB listed in Table 3, M2, M5 and M7 show better skills in
capturing WS10. In addition, the multi-model ensemble mean show the lowest RMSE for Q2,
and also better skills than most models for T2 and WS10.  The correlation coefficients between




multi-model ensemble mean and observations are 0.99, 0.99 and 0.98 for T2, Q2 and WS10,
respectively.
The accuracy of radiation predictions is of great significance in evaluating aerosol-radiation-
weather interactions. We evaluated simulated daily maximum SWDOWN averaged over sites in
northern China and southern China separately in January 2010 against observations. The
locations of the radiation sites are shown in Figure S5. As shown in Figure 5 (d), over stations in
northern China, all models except M6 and M7 reproduce daily maximum SWDOWN well, with
correlation coefficients ranging from 0.72 to 0.94. SWDOWN decreases under conditions of
high PM, as shown for example on January 9 and 15-21. This is one of the important reasons for
coupled air quality and meteorology modeling, as they can account for this effect of aerosols. It
is worth noting that most models predict higher daily maximum SWDOWN compared to
observations when severe haze happened in the North China Plain (16-19 January 2010),
indicating aerosol effects on radiation might be underestimated. Over southern China sites
(Figure 5e), M6 and M7 show a better consistence with observations than over northern China
sites. According to the calculated RMSE listed in Table 3, M3 and multi-model ensemble mean
exhibit relatively better performance in capturing the observed time series of daily maximum
SWDOWN in both northern China and southern China.
The above comparisons show that T2 and Q2 are reproduced well by the participating models,
and WS10 is overestimated by all models. Emery et al. (2001) proposed that excellent model
performance would be classified as wind speed RMSE smaller than 2 m/s, and wind speed bias
smaller than 0.5 m/s. Based on the calculated RMSE and MBE of WS10 shown in Table 3,
RMSE values from all models match the proposed RMSE threshold but MBE values are higher
than 0.5 m/s. The vertical distributions of temperature, water vapor mixing ratio and wind speeds



were also validated against atmospheric sounding data in Beijing at 1km and 3km (Figure S7,
averaged at 00:00 and 12:00 UTC) (http://weather.uwyo.edu/upperair/sounding.html). The
magnitudes of temperature, water vapor mixing ratio and wind speeds from different models are
generally consistent with each other at 1km and 3km, but variations are larger near surface.

**4.2 Evaluation of air pollutants**

Figure 6 displays the daily averaged predicted and observed $SO_2$, $NO_x$, CO, $O_3$, $PM_{2.5}$, and $PM_{10}$
at the Beijing station, along with the observation standard deviation (locations are shown in
Figure S6). Comparisons for the Tianjin, Shijiazhuang and Xianghe sites are shown in Figure S8-
S10. M6 only provided $SO_2$, $NO_x$ concentrations, so it is not only shown in the plots of CO, $O_3$,
$PM_{2.5}$, and $PM_{10}$. The observed and predicted primary pollutants and $PM_{2.5}$ and $PM_{10}$ show the
same monthly variations with elevated values at roughly weekly intervals, with the largest event
occurring during January 15-21. For example, as shown in the comparisons of $SO_2$
concentration, the temporal variations are reproduced well by all the models, but peak values are
overestimated or underestimated by some models. Based on the calculated MBE values shown in
Table 4, all models except M2 tend to underestimate $SO_2$ in Beijing. M1 shows the highest
correlation (0.90) with $SO_2$ observations in the Beijing site, and most other models show similar
good correlations. The multi-model ensemble mean shows a better agreement with observations
with a higher correlation of 0.92, and it falls within the range shown with standard deviation
error bar. In general, the predictions for $NO_x$ capture the main features in the observations, with
slightly less skill than for the $SO_2$ prediction. The calculated correlation coefficients for $NO_x$
from different models are close to each other, ranging from 0.63 to 0.88. M2 and M5 predict
higher $NO_x$ concentrations than observations and other models (MBE in Table 4). All models
overestimate $NO_x$ concentration in Shijiazhuang (Figure S8), suggesting $NO_x$ emissions in



Shijiazhuang might be overestimated in the MIX emission inventory. All models are consistent
with each in CO predictions.
PM$_{2.5}$ concentrations are well modelled, with high correlation coefficients ranging from 0.87 to
0.90 in Beijing, from 0.83 to 0.93 in Tianjin, and from 0.74 to 0.91 in Xianghe. The correlation
coefficient of the multi-model ensemble mean for PM$_{2.5}$ reaches 0.94 (Table 4), better than any
individual model. The performances of all participating models in reproducing PM$_{10}$ variations
are not as good as reproducing PM$_{2.5}$. M1 and M2 overestimate PM$_{10}$ concentrations, and other
models underestimate PM$_{10}$ concentrations (MBE in Table 4). These biases are probably related
to different treatments of primary aerosols and anthropogenic dust in the models.
The models showed the poorest skill in predicting ozone. All models exhibit different
performances in simulating ozone concentrations, and the correlation coefficients between
models and observations can reach negative values (Figure S8). M3 and M4 tend to overestimate
ozone concentrations, M2 slightly overestimates it, and M1, M5, and M7 slightly underestimate
it (MBE in Table 4). According the calculated RMSE in Table 4, M1 and M7 shows relatively
better performance in modeling ozone variations. Although WRF-Chem and NU-WRF models
were applied at three institutions, different gas phase chemistry schemes were used, which leads
to these diversities among predicted ozone concentrations. The impacts of gas phase chemical
mechanisms on ozone simulations have been investigated in Zhang et al. (2012); but under high
photochemical conditions. The results presented here winter conditions with slower
photochemistry in general and where hazy conditions further reduce photochemistry through
diming effects.



Figure 7 shows the comparisons between modeled and observed ground level daily averaged
concentrations of $SO_2$, $NO_x$, $O_3$ and $PM_{10}$ during January 2010 at the Rishiri site in Japan from
EANET. The locations of used EANET sites are marked in Figure S6. Comparisons at other
EANET sites are shown in Figure S11-S14. The models are able to predict the major features in
the observations. For example, low values of most pollutants are observed (and predicted) during
the first half of the month, followed by elevated values, which peak on January 21. For $SO_2$,
most models show similar capability in producing the temporal variations in observations with
slight underestimation (MBE in Table 5). According to the calculated RMSE averaged over all
the EANET sites, M2 and the multi-model ensemble mean performed the best. For $NO_x$, the
multi-model ensemble mean shows lower RMSE than any individual model (Table 5). Similar to
the comparisons over CARE-China sites, large discrepancies exist in ozone predictions, but the
model ensemble mean still shows lowest RMSE for ozone predictions. $PM_{10}$ concentrations are
largely underestimated by M1 (largest negative MBE: -21.03ug/m$^3$) and overestimated by M5
(highest positive MBE: 3.77ug/m$^3$) (Table 5), which could also be related to differences in the
way sea-salt emissions are treated in the various models.
**4.3 $PM_{2.5}$ and $PM_{2.5}$ chemical composition distribution**
Haze pollution is characterized by high loadings of $PM_{2.5}$, thus accurately predicting $PM_{2.5}$ and
its chemical compositions are crucial to understand haze pollution and to provide insightful
implications for controlling haze in China. The accuracy of predicting $PM_{2.5}$ chemical
composition is also of great importance in estimating aerosol-radiation interactions. For example,
black carbon absorbs shortwave radiation, whereas sulfate and organic carbon mostly scatter
radiation. Due to different implementations of chemical reactions in the models, predicted $PM_{2.5}$
chemical compositions from participating models differ largely. Figure 8 and Figure 9 show the



predicted monthly mean concentrations of sulfate, nitrate, ammonium, BC and OC in PM$_{2.5}$ from
all participating models for January 2010.
M1, M2, M3, M4 and M7 all predict quite low sulfate concentrations in east China, but with
considerably enhanced sulfate in southwest areas of China and west areas of India. M5 and M6
shows similar spatial patterns of sulfate except that M6 produces higher concentrations. The
chemical production of sulfate is mainly from gas-phase oxidation of SO$_2$ by OH radicals and
aqueous-phase pathways in cloud water. In cloud water, dissolved SO$_2$ can be oxidized by O$_3$,
H$_2$O$_2$, Fe(III), Mn(II), and NO$_2$ (Seinfeld and Pandis, 2016). Most chemical transport models have
included the above gas phase oxidation of SO$_2$ by OH and oxidation of SO$_2$ by O$_3$ and H$_2$O$_2$ in
aqueous phase. Under hazy conditions, radiation is largely reduced due to aerosol dimming effects,
and sulfate formation from gas phase and aqueous phase oxidation processes are slowed down,
which tend to reduce sulfate concentration. However, field observations exhibit an increase in
sulfate concentration during haze episode (Zheng et al., 2015), and Cheng et al. (2016) proposed
that the reactive nitrogen chemistry in aerosol water could contribute significantly to the sulfate
increase due to enhanced sulfate production rates of NO$_2$ reaction pathway under high aerosol pH
and elevated NO$_2$ concentrations in the North China Plain (NCP) during haze periods. Wang et al.
(2016) also pointed out the aqueous oxidation of SO$_2$ by NO$_2$ is key to efficient sulfate formation
on fine aerosols with high relative humidity and NH$_3$ neutralization or under cloudy conditions.
Besides, Zheng et al. (2015) suggested that heterogeneous chemistry on primary aerosols could
play an important role in sulfate production and lead to increasing sulfate simulation during haze
episodes. The above aqueous and heterogeneous processes are currently not incorporated in the
participating models for this study, which might be responsible for the apparent under-predictions



of sulfate concentration (Figure 10). M5 also incorporated heterogeneous chemical reactions on
aerosol surface (Li and Han, 2010), which enhances total sulfate production.
M1 and M5 predict relatively small nitrate and ammonium concentrations; while M2, M6 and
M7 produces similar magnitudes and spatial patterns of nitrate. Nitrate formation involves both
daytime and nighttime chemistry. During daytime, $NO_2$ can be oxidized by OH to form nitric
acid ($HNO_3$), and by ozone to form $NO_3$. $HNO_3$ is easily removed by dry or wet deposition, but
$NO_3$ is easily photolyzed back to $NO_2$. During nighttime, $NO_3$ is the major oxidant, which oxides
$NO_2$ to form dinitrogen pentoxide ($N_2O_5$). Homogenous reaction of $N_2O_5$ with water vapor is
possible but very slow while heterogeneous uptake of $N_2O_5$ onto aerosol particles has been
identified as a major sink of $N_2O_5$ and an important contributor to particulate nitrate (Kim et al.,
2014). The MOSAIC aerosol module (Zaveri et al., 2008) coupled with CBMZ gas phase
chemistry in WRF-Chem already includs heterogeneous uptake of $N_2O_5$ since version v3.5.1
(Archer-Nicholls et al., 2014), which is the version used by M2, leading to the high production of
nitrate. M7 also predict high nitrate concentrations, and the predicted lower nitrate
concentrations from other models are probably due to missing aqueous phase and heterogeneous
chemistry. M3 and M4 do not include the explicit nitrate and ammonium treatment but
ammonium is implicitly considered in total $PM_{2.5}$ mass estimate.
The predicted ammonium concentrations are associated with the amounts of sulfate and nitrate,
as shown by its similar spatial distribution to sulfate and nitrate. $NH_3$ neutralizes $H_2SO_4$ and
$HNO_3$ to form aerosol, so its amount can affect the formation of sulfate, nitrate and ammonium.
Since the same emission inventory was used, the amount of ammonia available for neutralizing
will not vary greatly among these models. Thus, the rates of $H_2SO_4$ and $HNO_3$ production
determines the amounts of ammonium. For example, the produced ammonium concentrations are



small in M1, similar to their sulfate and nitrate productions. High ammonium concentrations are
predicted from M6, due to high productions of nitrate and sulfate (Figure 8).
The spatial distributions and magnitudes of predicted BC from all participating models are
similar to each other as BC is a primary pollutant whose mass as BC is not impacted by chemical
reactions. The concentrations of BC in the atmosphere are mainly influenced by PBL mixing and
diffusion, aging, deposition and advection. Predicted BC from M2 and M7 are higher than from
other models, which might be caused by different treatments of emission inventory (for example,
how to distribute emissions to different vertical layers), horizontal grid interpolation, and/or
different parameterizations for vertical diffusion, aging, deposition and advection.
The disparity among predicted OC concentrations is mainly associated with the different
treatments of SOA production, given the POC prediction is generally consistent among models
using the same emission inventory. The predicted OC concentrations from M1, M2, and M7 are
close to each other. M1 uses SORGAM (Secondary Organic Aerosol Model) to simulate SOA, but
M2 and M6 did not include any SOA formation mechanism. The similar magnitudes of OC from
M1 and M2 suggest that SORGAM in M1 does not produce appreciable amounts of SOA, which
is consistent with the findings in Gao et al. (2016a). Although SOA formation is implemented in
M5, the production is relatively weak compared to M3 and M4. In the atmosphere, SOA is mainly
formed from the condensation of semi-VOCs from oxidation of primary VOCs. An empirical 2-
produt model (Odum et al., 1996) is often used to simulate SOA formation, but this method was
reported to significantly underestimate measured SOA mass concentrations. Later, the volatility
basis-set approach (Donahue et al., 2006) was developed to represent more realistically the wide
range of volatility of organic compounds and more complex processes, and it was found to increase
SOA production and to reduce observation-simulation biases in many regions with high emissions



(Tsimpidi et al., 2010) including east China (Han et al., 2016). It was also suggested that primary
organic aerosols (POA) are semi-volatile and can evaporate to become SOA precursors, which
promotes the understanding and improvements of SOA modeling (Li et al., 2011). In M5, the SOA
production is calculated using a bulk yield method via Lack et al. (2004), in which the amount of
SOA able to be produced from a unit of reacted VOC from anthropogenic and biogenic  origins
are  used to represent SOA yields. However, the SOA concentration is highly dependent on the
yield data. During haze episodes, photochemistry is reduced due to the aerosol dimming effect,
thus aqueous reaction processes on aerosol water and cloud/fog water could become much more
important in producing SOA as suggested in Cheng et al. (2016). The missing representation of
such process in the participating models may partly account for the low values in the simulated
SOA. In M3 and M4, SOA is treated by assuming that 10% of VOCs from terrestrial source are
converted to OC (Chin et al., 2002), and these models produce high OC concentrations, with a
major contribution from SOA. The 10% yield rate could be unrealistically high during hazy days
because solar radiation was much reduced.
The different predictions of $PM_{2.5}$ chemical components lead to differences in $PM_{2.5}$ concentrations
for January 2010, which are shown in the last row of Figure 9. Although spatial distributions of
$PM_{2.5}$ from these models are similar, the underlying causes are different. M2, M3 and M5 simulated
higher $PM_{2.5}$ levels in deserts of west China, which are contributed by dust deflation. M1 and M7
fail to produce high $PM_{2.5}$ concentrations in the deserts of west China, due to omission of dust
emissions. M4 presented results in a smaller domain excluding west China. The enhanced $PM_{2.5}$
concentrations in Central China from M2 and M7 are caused by large nitrate production, as shown
in Figure 8.



The differences in the predictions of aerosols composition discussed above can be seen clearly in
the comparisons at the Beijing site on 13-23 January when a haze event occurred in the NCP
(Figure 10). Also shown are the observed values. Most models fail to produce the observed high
sulfate concentrations. Only M5 prediction is close to observation, and M6 predicts higher
sulfate level. M2 and M7 predict reasonable nitrate concentrations. M3 and M4 largely
overpredict OC during haze period, but other models tend to underpredict OC concentrations.
Figure 11 and 12 show the ensemble mean monthly averaged near-surface $PM_{2.5}$, $PM_{2.5}$
composition, along with the spatial distribution of the coefficient of variation. The coefficient of
variation is defined as the standard deviation divided by the average (Carmichael et al., 2008), and
larger values indicate lower consistency among models. Mean concentrations of $PM_{2.5}$ and $PM_{2.5}$
chemical compositions are high in Sichuan Basin and east China. High coefficient of variation are
shown in North China for sulfate, and in most areas for nitrate and OC. The diversity in predictions
of these species are caused by complexity of secondary formation and different treatments in
models as discussed earlier. Higher consistency is shown for model BC with coefficient of
variations less than 0.3 in most areas (Figure (h)). Coefficient of variations for $PM_{2.5}$ are also low
in North China region, which is consistent with good performance of $PM_{2.5}$ predictions shown in
above comparisons. However, the coefficient of variation can reach above 1.6 in northwestern
regions, partially due to discrepancies in dust predictions.
**4.4 Evaluation of AOD**
AOD is an indication of aerosol pollution, which tells us how much sunlight is blocked from
reaching the surface by suspended aerosols. We used the measurements of AOD at AERONET
and CARE-China sites to evaluate how participating models perform in simulating AOD. In



WRF-Chem, AOD is usually calculated at 300, 400, 600 and 999nm, which can be converted to
AOD at other wavelengths based on Angstrom exponent relation (Schuster et al., 2006). The
submitted AOD from all models except M6 are at 550nm, and AOD from M6 are at 495nm. We
used Angstrom exponent relation (Schuster et al., 2006) to convert AOD from M6 at 495nm to
550nm, and all used AERONET and CARE-China AOD data to 550nm. The locations of
AERONET and CARE-China AOD measurement sites are shown in Figure S5. Daytime mean
AOD are calculated in pairwise manner and the comparisons and performance statistics are
shown in Figure 13, 14, and Table 6.  On some days, data are missing because AOD cannot be
retrieved under serious pollution conditions (Gao et al., 2016a). On days with data, the variations
of AOD are captured well by all models. However, large disparities exist among models in the
simulated peak AOD values (factor of 2) at monitoring stations during the severe haze episode
on 15-20 January 2010 (Figure 13 and Figure 14).  The participating models exhibit various skill
in simulating AOD temporal variation at different sites.
At CARE-China sites, M7 produces the best correlation coefficient R (0.83) among models at
Baoding and Beijing forest sites, M2 produces the highest R (0.86) at Cangzhou site, whereas
M5 shows the highest R (0.93) at the Beijing city site.  At AERONET sites, M7 shows the
highest R (0.81) at Beijing, whereas M2 and M5 produce R as high as 0.91 at Xianghe site,
which is about 60km southeast of downtown Beijing. In terms of AOD magnitude, it's interesting
to note that during the severest haze days around 19 January 2010, M2 consistently simulates the
highest AOD among models, followed by M5 and M7, with the lowest AOD from M6, and other
models in the middle at the sites (Baoding, Beijing City, Beijing Forest, Cangzhou, Beijing,
Xianghe) in the north China plain (NCP).  It is important to explore the causes for the disparities
in AOD predicitons.



AOD is calculated as the vertical integration of extinction coefficient, which is a function of
particle mass extinction efficiency (extinction cross section) and mass concentration. The
extinction efficiency is determined by particle size, refractive index and wave length. Aerosol size
can grow bigger as ambient relative humidity increases, which is known as aerosol hygroscopic
growth. The overall extinction coefficient of all aerosols also depends on mixing state among
aerosols. Therefore, AOD is closely related a series of affecting factors from both aerosol physical
properties, mass concentration and meteorological conditions.
In M1, M5, M6 and M7, particle size distribution is described by a lognormal function with a
geometric mean radius and a geometric standard deviation basically based on OPAC (Optical
properties of aerosols and clouds) database (Hess et al. 1998). In M3 and M4, sulfate, BC and OC
are parameterized in bulk mode, and a sectional scheme is used for sea-salt and dust aerosols. M2
uses an 8 bins sectional aerosol scheme with size sections ranging from 39nm to 10μm. The
refractive index of various aerosol components in the models are mainly taken from d'Almeida et
al. (1991) or OPAC database. All models except M6 use a kappa (κ) parameterization (Petters and
Kreidenweis, 2007), in which the aerosol hygroscopicity κ largely varies among different aerosol
chemical components, such as κ=0 for black carbon, and κ>0.6 for inorganic aerosols, but the
prescribed κ values could be different in the above models. M6 uses a different hygroscopic growth
scheme following Kiehl and Briegleb (1993). WRF-Chem models assume internally mixing
among aerosols within each mode and externally mixing between modes, M5 assumes inorganic
and carbonaceous aerosols are internally mixed and externally mixed with soil dust and sea-salt.
M6 uses an external mixture assumption among aerosols except for hydrophilic BC, which is
internally mixed with other aerosols in a core-shell way.



We first look at the mass concentrations of different aerosol components because of their important
roles in determining optical properties. The observed total inorganic aerosol concentration in
Beijing on 19 January 2010 is about $130\mu g/m^3$, with sulfate and nitrate being about 50 and $65\mu g/m^3$,
respectively (Figure 10). The models generally predict a much lower sulfate concentration variation except
that the prediction from M5, which is close to observations, and M6, which shows overprediction.
Most models predict lower nitrate concentration, in contrast to the overprediction by M2. In terms
of inorganic aerosols, which have a similar optical properties, the total concentration (the sum of
sulfate, nitrate and ammonium) from M2 ($175\mu g/m^3$) is higher than observation and other models,
and this can explain the largest simulated AOD by M2. M6 simulates a similar level of inorganic
aerosols to M2, but the simulated AOD is lower than other models, which could be due to a weak
hygroscopicity or lower RH simulation (see Figure S14). For example, high RH on January 19 are
captured by M2 and M6, but underpredicted by M6 (Figure S14a). Although M3 and M4 largely
overpredict OC concentration, their simulated AOD are lower than M1 and M5 because their
simulated inorganic aerosol concentrations are much lower and OC has a smaller (mass) extinction
coefficient than inorganic aerosols. M1 predicts about three times larger BC concentration than
the observations, although the mass extinction coefficient of BC is even larger than inorganic
aerosols, the mass concentration and hygroscopicity of BC are much smaller and weaker than that
of inorganic aerosols, leading to relatively lower AOD from M1 simulation. M5 and M7 predict a
similar level of inorganic aerosol concentrations ($80{\sim}90\mu g/m^3$) and use a similar hygroscopic
growth scheme, and this can help explain their consistency in the simulated AOD magnitude. In
general, it appears the magnitude of inorganic aerosol concentrations and the hygroscopic growth
efficiency (affected by varied RH) can account for or explain the simulated variations and



magnitudes of AOD in Beijing during the severe haze event, given the aerosol size distribution
and mixing state are alike among models.
Table 6 shows the statistics for AOD simulation at NCP sites and at all sites. In the NCP region,
R ranges from 0.36~0.74 for all the models. M2, M5 and M7 produce R around 0.7, indicating a
better simulation of AOD variations. M2 and M7 exhibit the best R (0.65) for all sites. It's
noteworthy that R values at the sites in NCP are larger than that at all sites, indicating the larger
reliability of model inputs (emissions and boundary conditions) and meteorological simulations.
In terms of magnitudes, all models tend to underpredict AOD in the whole domain, with NMB of
-2.7 to -71% in the NCP, and larger biases (NMB of -21~-75%) at all sites. M7 shows the smallest
MBE (-0.05) and NMB (-2.7%) and M2 produces the smallest RMSE. It is interesting to note that
the simulated AOD from the WRF-Chem models differed largely (-12 to -71%) between M1 and
M3 at the NCP sites, and the WRF-Chem model using finer grid size (M4) can produced slightly
smaller NMB compared with the same model using larger grid size (M3). However, as grid size
becomes finer, R and RMSE from M4 may become worse, although AOD magnitude improved.
The effect of grid resolution will be a topic of future paper.

## 5 Summary

The MICS-Asia Phase III Topic 3 examines how current online coupled air quality models
perform in reproducing extreme aerosol pollution episodes in North China, and how high aerosol
loadings during these episodes interact with radiation and weather. Two hazy winter months,
namely January 2010 and January 2013, were studied by six modeling groups from China, Korea
and the United States. Predicted meteorological variables and air pollutants from these modeling





groups were compared against each other, and measurements as well. A new anthropogenic
emission inventory was developed for this phase (Li et al., 2017), and this inventory along with
biogenic, biomass burning, air and ship, and volcano emissions were provided to all modeling
groups. All modelling groups were required to submit results based on the analysis methodology
that documented in this paper.
Comparisons against daily meteorological variables demonstrate that all models can capture the
observed near surface temperature and water vapor mixing ratio, but near surface wind speeds
are overestimated by all models to varying degrees. The observed daily maximum downward
shortwave radiation, particularly low values during haze days, were represented in the
participating models. Comparisons with measurements of air pollutants, including $SO_2$, $NO_x$,
CO, $O_3$, $PM_{2.5}$, and $PM_{10}$, from CARE-China and EANET networks showed that the main
features of accumulations of air pollutants are represented in current generation of online
coupled air quality models. The variations in observed AOD from CARE-China and AERONET
networks were also reproduced by the participating models. Differences exist between simulated
air pollutants, particularly ozone, which are probably related to different treatments of emission
inventory, different meteorological and chemical parameterizations, and uncertainties in
interpolations from original emission inventory to model grids might also contribute to these
differences.
Manifold diversities were found in the predicted $PM_{2.5}$ chemical compositions, especially
secondary inorganic aerosols and organic carbon. During winter haze events, the production
from gas phase chemistry is inhibited, and whether including other aerosol formation pathways
(such as aqueous phase chemistry), or how these chemistry is parametrized leads to the large
difference between simulated concentrations of secondary inorganic aerosols. In addition,




differences in treatments of SOA also lead to large discrepancies between simulated OC
concentrations.
These results provide some directions for future model improvements, and underscore the
importance of accurately predicting aerosol concentration and compositions. Differences in the
simulated variations and magnitudes of AOD in Beijing during the severe haze event could be
explained by the differences in simulated inorganic aerosol concentrations and the hygroscopic
growth efficiency (affected by varied RH).
Previous studies have studied radiative forcing during haze event (Gao et al., 2017), but there are
large uncertainties in aerosol modeling during haze events and in estimating its interactions with
weather and climate. The uncertainties come from model inputs (land use data, model initial and
boundary conditions, etc.), physical and chemical mechanisms, and particularly
parameterizations of aerosol-radiation-weather interactions. Other companion papers with
respect to MICS-Asia Topic 3 will provide more insights into current knowledge of aerosol-
weather interactions under heavy pollution conditions.

**ACKNOWLEDGMENTS**
The authors would like to acknowledge support of this project from National Natural Science
Foundation of China (No. 91644217), and ground measurements from Yuesi Wang's research
group. The ground observation was supported by the National Natural Science Foundation of
China (41222033; 41375036) and the CAS Strategic Priority Research Program Grant
(XDA05100102, XDB05020103).



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



Table 1 Participating models in Topic 3

| Models | M1: WRF-Chem1 | M2: WRF-Chem2 | M3: NU-WRF1 | M4: NU-WRF2 | M5: RIEMS-Chem | M6: RegCCMS | M7: WRF-CMAQ |
|---|---|---|---|---|---|---|---|
| **Modelling Group** | Pusan National University | University of Iowa | USRA/NASA | USRA/NASA | Institute of Atmospheric Physics | Nanjing University | University of Tennessee |
| **Grid Resolution** | 45km | 50km | 45km | 15km | 60km | 50km | 45km |
| **Vertical Layers** | 40 layers to 50mb | 27 layers to 50mb | 60 layers to 20mb | 60 layers to 20mb | 16 layers to 100mb | 18 layers to 50mb | |
| **Gas phase chemistry** | RACM | CBMZ | RADM2 | RADM2 | CBM4 | CBM4 | SAPRC99 |
| **Aerosols** | MADE | MOSAIC-8bin | GOCART | GOCART | Sulfate, nitrate, ammonium, BC, OC, SOA, 5 bins of soil dust, and 5 bins of sea salt | Sulfate, nitrate, ammonium, BC and POC | AE06 |
| **Chemical Boundary Conditions** | Climatological data from NALROM | MOZART | MOZART GOCART | MOZART GOCART | GEOS-Chem | Climatological data | GEOS-Chem |



Table 2 CARE-Chine network sites

| ID | Site name | Characteristics | Longitude | Latitude |
|----|-----------|-----------------|-----------|----------|
| 1 | Beijing | AOD | 116.37 | 39.97 |
| 2 | Tianjin | Air quality* | 117.21 | 39.08 |
| 3 | Shijiazhuang | Air quality | 114.53 | 38.03 |
| 4 | Xianghe | Air quality | 116.96 | 39.75 |
| 5 | Xinglong | Air quality | 117.58 | 40.39 |
| 6 | Beijing Forest | AOD | 115.43 | 39.97 |
| 7 | Baoding | AOD | 115.51 | 38.87 |
| 8 | Cangzhou | AOD | 116.80 | 38.28 |
| 9 | Shenyang | AOD | 123.63 | 41.52 |
| 10 | Jiaozhou Bay | AOD | 120.18 | 35.90 |

*Air quality: surface $PM_{2.5}$, $PM_{10}$, $SO_2$, $NO_x$, CO, $O_3$



Table 3 Performance Statistics of Meteorology Variables (RMSE and MBE units: degree for T2; g/kg for Q2; m/s for WS10; W/m$^2$ for SWDOWN)

| Metrics | Models | T2 | Q2 | WS10 | SWDOWN South | SWDOWN North |
|---|---|---|---|---|---|---|
| RMSE | M1 | 0.64 | 0.14 | 2.04 | 86.32 | 69.39 |
| | M2 | 0.68 | 0.10 | 0.95 | 96.71 | 72.76 |
| | M3 | 2.34 | 0.16 | 1.16 | 60.34 | 59.56 |
| | M4 | 2.90 | 0.43 | 1.44 | 100.34 | 74.89 |
| | M5 | 2.97 | 0.46 | 0.91 | 91.06 | 65.27 |
| | M6 | 3.57 | 0.76 | 2.48 | 85.63 | 222.00 |
| | M7 | 2.05 | 0.17 | 0.22 | 158.10 | 218.67 |
| | Ensemble | 1.81 | 0.10 | 1.28 | 81.96 | 62.51 |
| MBE | M1 | -0.19 | 0.02 | 2.01 | 66.58 | 59.94 |
| | M2 | -0.60 | -0.01 | 0.91 | 83.88 | 62.38 |
| | M3 | -2.18 | -0.04 | 1.11 | 36.44 | 47.74 |
| | M4 | -2.09 | 0.11 | 1.40 | 26.78 | 33.59 |
| | M5 | -2.73 | 0.43 | 0.74 | 49.06 | 51.00 |
| | M6 | -3.06 | -0.56 | 2.37 | -0.49 | -202.26 |
| | M7 | -2.02 | -0.12 | 0.15 | 145.24 | 159.02 |
| | Ensemble | -1.71 | -0.02 | 1.25 | 65.54 | 36.37 |
| NMB (%) | M1 | -0.07% | 0.19% | 17.58% | 14.61% | 13.34% |
| | M2 | -0.21% | -0.12% | 7.94% | 18.41% | 13.88% |
| | M3 | -0.79% | -0.34% | 9.73% | 8.00% | 10.63% |
| | M4 | -0.75% | 0.95% | 12.26% | 5.88% | 7.48% |
| | M5 | -0.98% | 3.65% | 6.45% | 10.77% | 11.35% |
| | M6 | -1.10% | -4.77% | 20.73% | -0.11% | -45.02% |
| | M7 | -0.72% | -1.05% | 1.31% | 31.88% | 35.39% |
| | Ensemble | -0.61% | -0.14% | 10.98% | 14.38% | 8.10% |



Table 4 Performance Statistics of Air Pollutants at the CARE-China sites (RMSE and MBE units: ppbv for gases and µg/m$^3$ for PM)

| Metrics | Models | $SO_2$ | $NO_x$ | $O_3$ | $PM_{2.5}$ | $PM_{10}$ | | $SO_2$ | $NO_x$ | $O_3$ | $PM_{2.5}$ | $PM_{10}$ |
|---|---|---|---|---|---|---|---|---|---|---|---|---|
| **r** | **M1** | 0.76 | 0.60 | 0.46 | 0.85 | 0.76 | | -17.14 | -5.53 | -1.54 | 55.69 | 30.70 |
| | **M2** | 0.77 | 0.65 | 0.48 | 0.90 | 0.85 | | 2.10 | 33.41 | 2.53 | 48.44 | 12.94 |
| | **M3** | 0.69 | 0.66 | 0.39 | 0.85 | 0.68 | **MBE** | -15.89 | -8.00 | 23.93 | 8.13 | -19.92 |
| | **M4** | 0.67 | 0.61 | 0.42 | 0.88 | 0.73 | | -9.98 | 0.28 | 24.49 | 23.12 | -3.23 |
| | **M5** | 0.72 | 0.73 | 0.39 | 0.91 | 0.84 | | -9.69 | 64.29 | -5.30 | 1.68 | -52.49 |
| | **M6** | 0.62 | 0.48 | - | - | - | | -27.53 | -29.98 | - | - | - |
| | **M7** | 0.57 | 0.58 | 0.48 | 0.82 | 0.77 | | -25.56 | 7.85 | -3.09 | 43.59 | -21.00 |
| | **Ensemble** | 0.79 | 0.71 | 0.51 | 0.94 | 0.87 | | -14.81 | 8.90 | 6.84 | 30.11 | -8.83 |
| **RMSE** | **M1** | 27.63 | 33.51 | 6.40 | 73.37 | 79.06 | | -14.05 | -5.41 | 7.37 | 63.57 | 18.93 |
| | **M2** | 21.00 | 66.30 | 8.15 | 72.44 | 80.72 | | 12.13 | 69.58 | 39.87 | 54.07 | 6.38 |
| | **M3** | 29.50 | 36.87 | 24.76 | 47.20 | 78.21 | **NMB** | -10.44 | -6.26 | 306.33 | 9.67 | -12.41 |
| | **M4** | 26.86 | 36.10 | 25.34 | 49.13 | 72.25 | **(%)** | 0.31 | 4.51 | 316.99 | 27.03 | -1.78 |
| | **M5** | 32.17 | 87.48 | 7.90 | 45.32 | 81.00 | | 6.83 | 127.45 | -38.49 | 0.52 | -32.94 |
| | **M6** | 33.95 | 48.62 | - | - | - | | -51.28 | -48.59 | - | - | - |
| | **M7** | 34.75 | 35.88 | 6.89 | 64.25 | 70.19 | | -37.87 | 18.32 | -7.78 | 48.92 | -12.78 |
| | **Ensemble** | 24.10 | 29.12 | 8.86 | 45.25 | 56.65 | | -13.48 | 22.80 | 104.04 | 33.96 | -5.77 |
| **MFB (%)** | **M1** | -17.32 | 5.26 | -5.06 | 64.34 | 21.98 | | 53.73 | 43.79 | 54.54 | 69.92 | 41.95 |
| | **M2** | 9.09 | 32.82 | 19.88 | 51.18 | 3.44 | | 43.18 | 73.39 | 60.79 | 59.87 | 39.35 |
| | **M3** | -12.96 | 4.52 | 113.60 | 32.67 | -4.62 | **MFE** | 57.87 | 46.69 | 113.60 | 50.10 | 36.83 |
| | **M4** | 1.53 | 15.34 | 114.35 | 45.27 | 6.07 | **(%)** | 46.30 | 48.13 | 114.35 | 55.03 | 34.72 |
| | **M5** | -20.24 | 67.25 | -62.65 | 16.88 | -35.15 | | 63.69 | 72.07 | 80.92 | 48.17 | 45.09 |
| | **M6** | -77.13 | -56.89 | - | - | - | | 84.21 | 69.66 | - | - | - |
| | **M7** | -46.67 | 21.80 | -19.50 | 57.19 | -7.02 | | 72.35 | 49.18 | 60.64 | 66.27 | 35.83 |
| | **Ensemble** | -14.17 | 26.41 | 62.86 | 50.61 | 3.12 | | 43.13 | 42.94 | 71.14 | 55.86 | 28.05 |





Table 5 Performance Statistics of Air Pollutants at the EANET sites (RMSE and MBE units: ppbv for gases and $\mu g/m^3$ for PM)

| Metrics | Models | $SO_2$ | $NO_x$ | $O_3$ | $PM_{10}$ | | $SO_2$ | $NO_x$ | $O_3$ | $PM_{10}$ |
|---|---|---|---|---|---|---|---|---|---|---|
| | M1 | 0.57 | 0.64 | 0.14 | 0.59 | | -0.68 | 0.68 | -6.16 | -21.03 |
| | M2 | 0.59 | 0.45 | 0.30 | 0.75 | | -0.45 | -0.39 | 5.50 | 3.12 |
| | M3 | 0.50 | 0.55 | 0.26 | 0.51 | | -0.37 | -0.21 | 3.67 | 3.55 |
| | M4 | 0.45 | 0.55 | 0.25 | 0.49 | | -0.57 | -0.61 | 4.28 | 2.96 |
| | M5 | 0.58 | 0.54 | 0.01 | 0.03 | | -0.57 | 1.28 | 4.67 | 3.77 |
| r | M6 | 0.33 | 0.24 | - | - | MBE | 0.32 | -1.68 | - | - |
| | M7 | 0.53 | 0.49 | 0.38 | 0.55 | | -0.03 | 0.64 | -1.89 | -15.75 |
| | Ensemble | 0.60 | 0.66 | 0.32 | 0.59 | | -0.34 | -0.07 | 1.68 | -3.89 |
| | M1 | -46.45 | 41.49 | -15.03 | -82.29 | | 1.18 | 1.37 | 8.23 | 23.39 |
| | M2 | -29.64 | -29.75 | 13.47 | 18.90 | | 1.01 | 1.35 | 7.29 | 10.01 |
| | M3 | -25.42 | -17.75 | 9.01 | 19.46 | | 1.02 | 1.02 | 6.44 | 13.71 |
| NMB | M4 | -39.63 | -35.84 | 10.47 | 16.95 | RMSE | 1.14 | 0.97 | 6.35 | 13.78 |
| (%) | M5 | -34.23 | 38.50 | 11.38 | 31.80 | | 1.27 | 2.75 | 12.27 | 23.10 |
| | M6 | 12.63 | -93.57 | - | - | | 1.38 | 1.85 | - | - |





| | | | | | | | | |
|---|---|---|---|---|---|---|---|---|
| **M7** | 17.42 | 31.47 | -4.71 | -56.18 | 1.04 | 1.57 | 6.52 | 18.76 |
| **Ensemble** | -20.76 | -10.79 | 4.10 | -8.56 | 0.96 | 0.79 | 4.98 | 11.69 |



Table 6 Performance Statistics of AOD

| Metrics | Models | M1 | M2 | M3 | M4 | M5 | M6 | M7 | Ensemble |
|---|---|---|---|---|---|---|---|---|---|
| R | North China | 0.63 | 0.74 | 0.57 | 0.51 | 0.68 | 0.36 | 0.71 | 0.77 |
| | All | 0.60 | 0.65 | 0.46 | 0.42 | 0.53 | 0.33 | 0.64 | 0.75 |
| MBE | North China | -0.25 | -0.10 | -0.09 | -0.07 | -0.13 | -0.21 | -0.05 | -0.03 |
| | All | -0.18 | -0.02 | -0.01 | -0.01 | -0.01 | -0.11 | 0.00 | -0.12 |
| NMB (%) | North China | -71.25 | -23.28 | -12.63 | -9.59 | -28.34 | -59.19 | -2.70 | -30.17 |
| | All | -74.94 | -30.69 | -25.68 | -23.64 | -28.24 | -55.38 | -21.12 | -28.91 |
| RMSE | North China | 0.35 | 0.20 | 0.26 | 0.28 | 0.24 | 0.36 | 0.22 | 0.22 |
| | All | 1.16 | 1.13 | 1.15 | 1.15 | 1.15 | 1.17 | 1.14 | 0.20 |

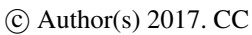



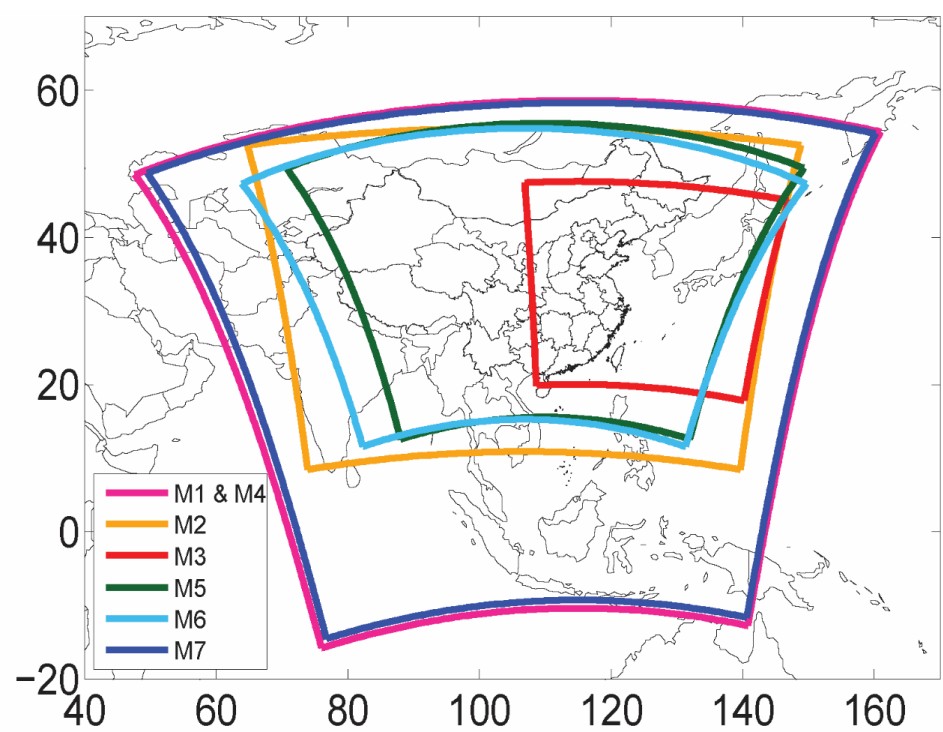

Figure 1. MICS-ASIA III Topic 3 modeling domains (descriptions of each model are

documented in Table 1)





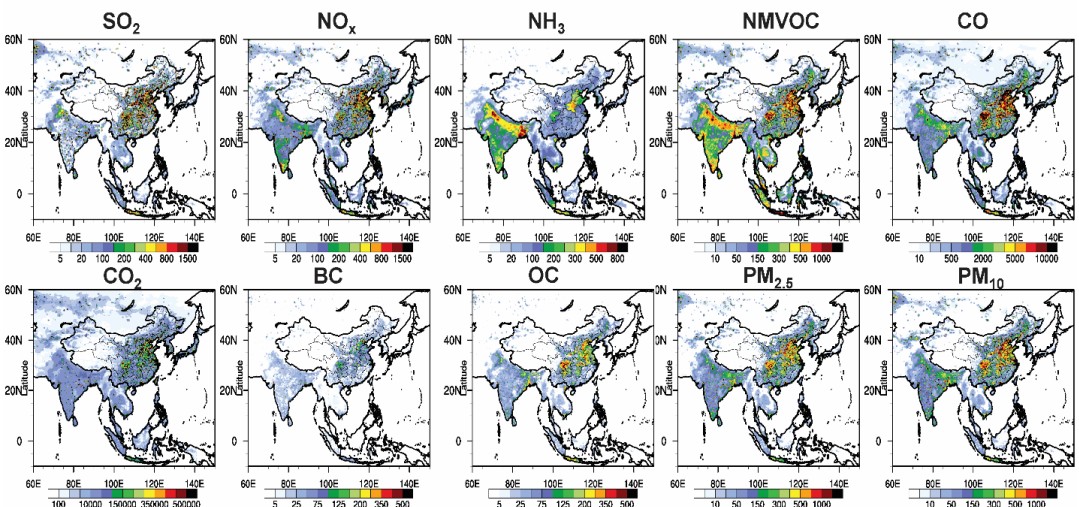

Figure 2. MIX emission inventory for January 2010 (Mg/month/grid)

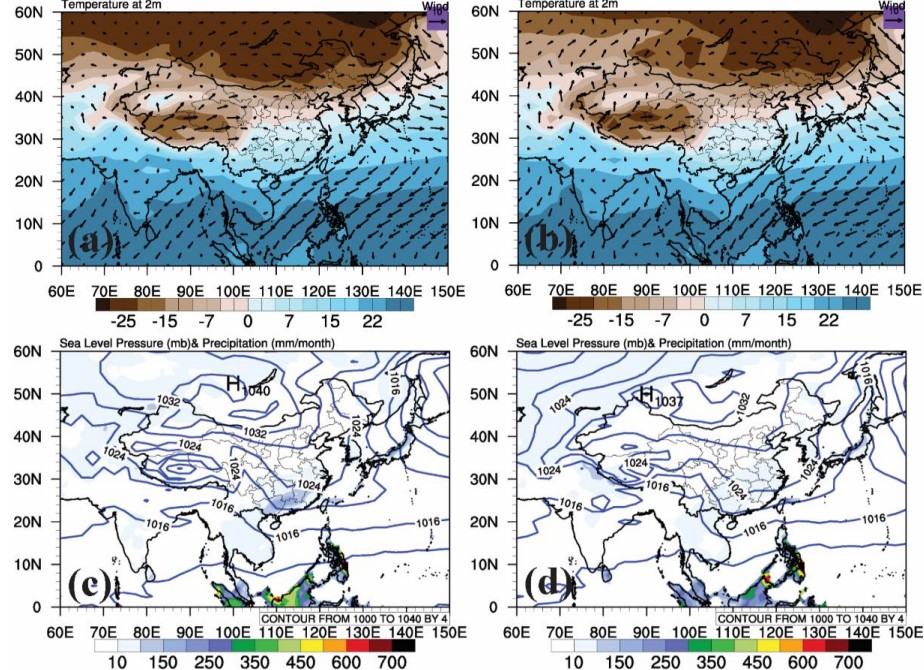

Figure 3. Monthly mean temperature at 2m, winds at 10m, total precipitation and sea level

pressure for January 2010 (a,c) and January 2013 (b,d)




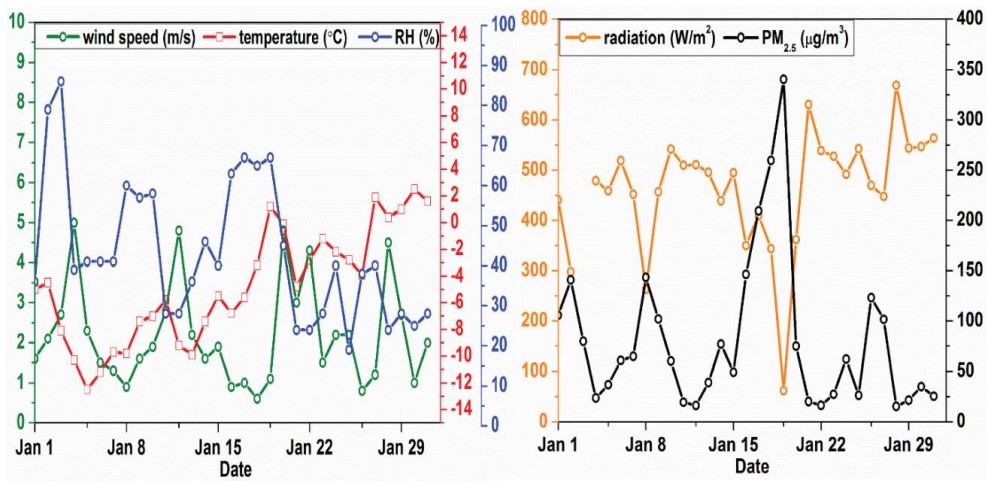

Figure 4. Observed near surface daily meteorological variables and PM$_{2.5}$ concentrations in

Beijing for January 2010

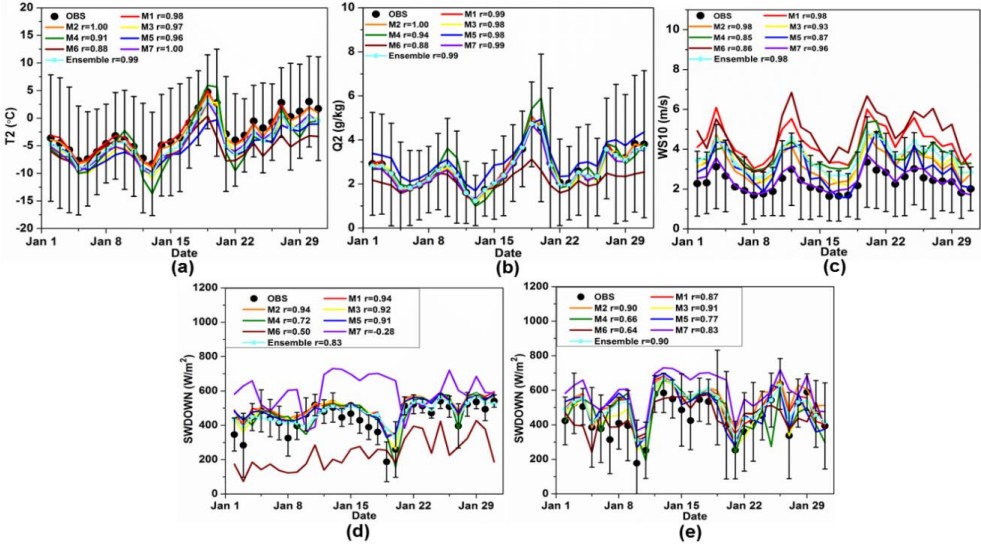

Figure 5. Comparisons between simulated and observed near surface temperature (a), water

vapor mixing ratio (b), and wind speeds (c) (T2, Q2, and WS10), downward shortwave radiation

in North China (d) and South China (e)

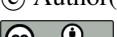



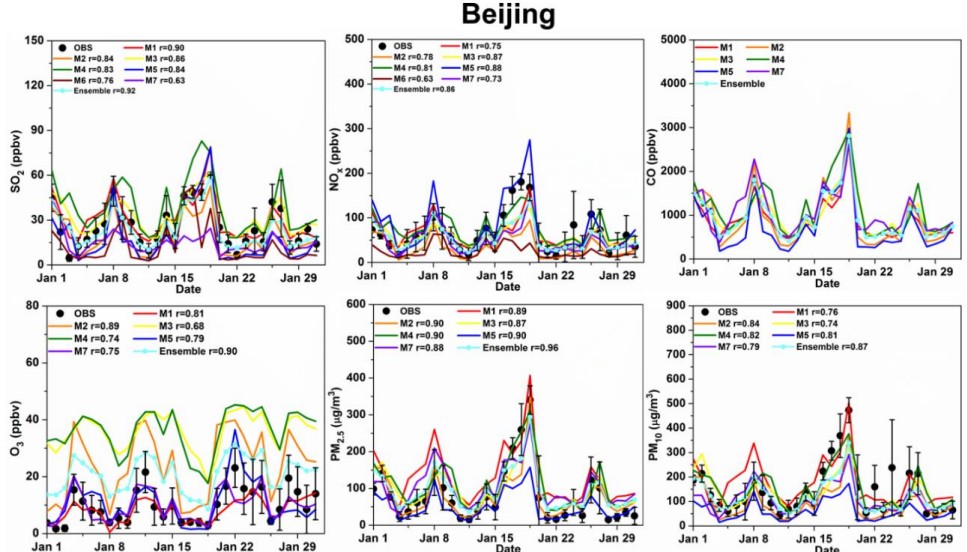

Figure 6. Comparisons between simulated and observed daily air pollutants (SO$_2$, NO$_x$, CO, O$_3$,

PM$_{2.5}$ and PM$_{10}$) at the Beijing CARE-China site

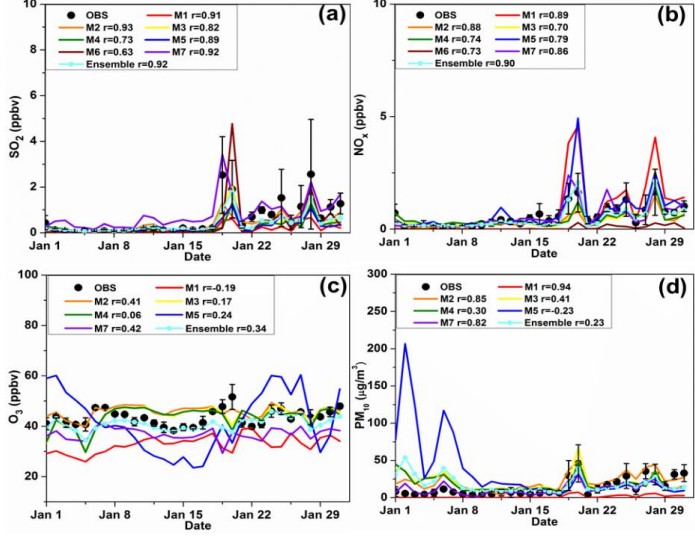

Figure 7. Comparisons between simulated and observed daily air pollutants (SO$_2$, NO$_x$, O$_3$, and

PM$_{10}$) at the Rishiri EANET sites



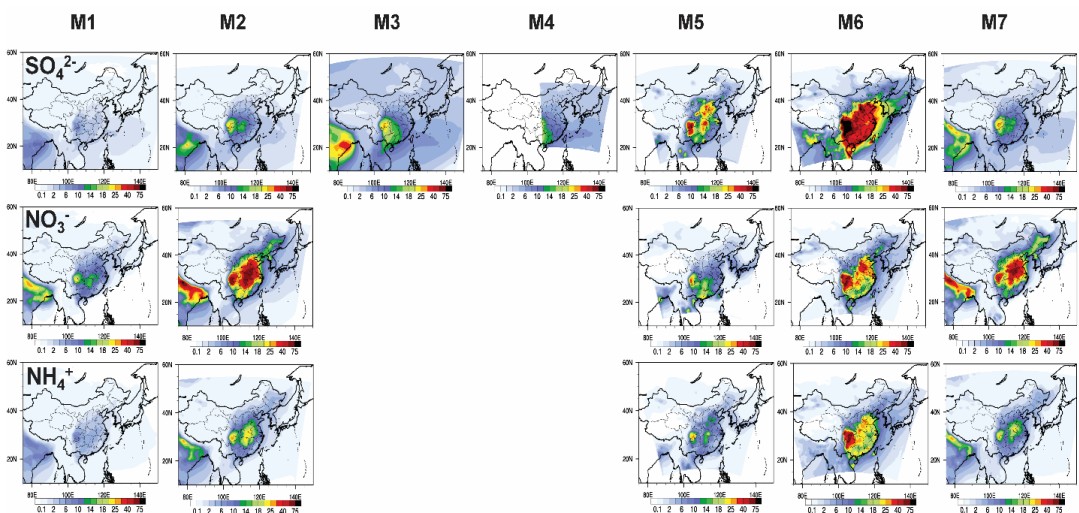

Figure 8. Simulated monthly concentrations of major PM$_{2.5}$ components ($\mu$g/m$^3$) for January

2010 from all participating models

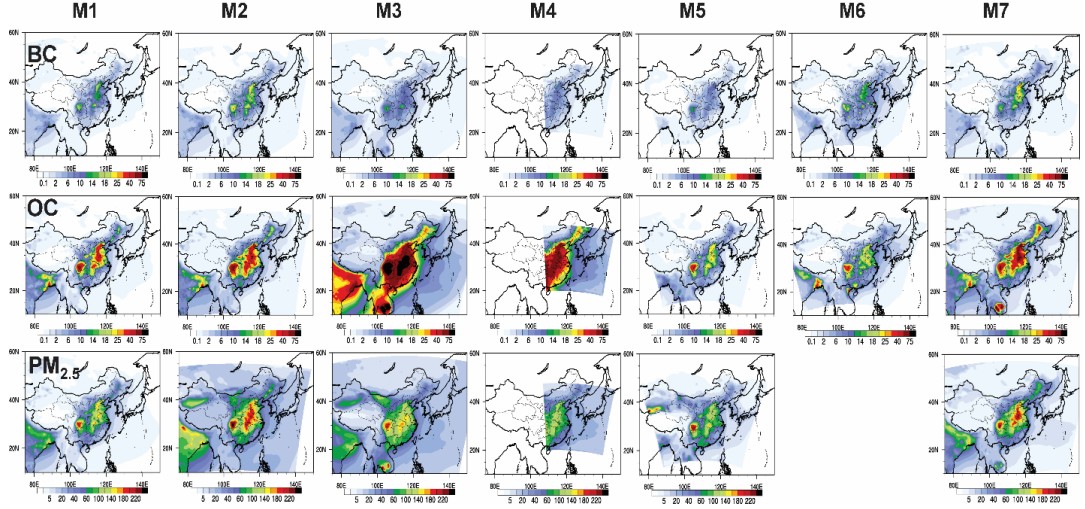

Figure 9. Simulated monthly concentrations of PM$_{2.5}$ and major PM$_{2.5}$ components ($\mu$g/m$^3$) for

January 2010 from all participating models




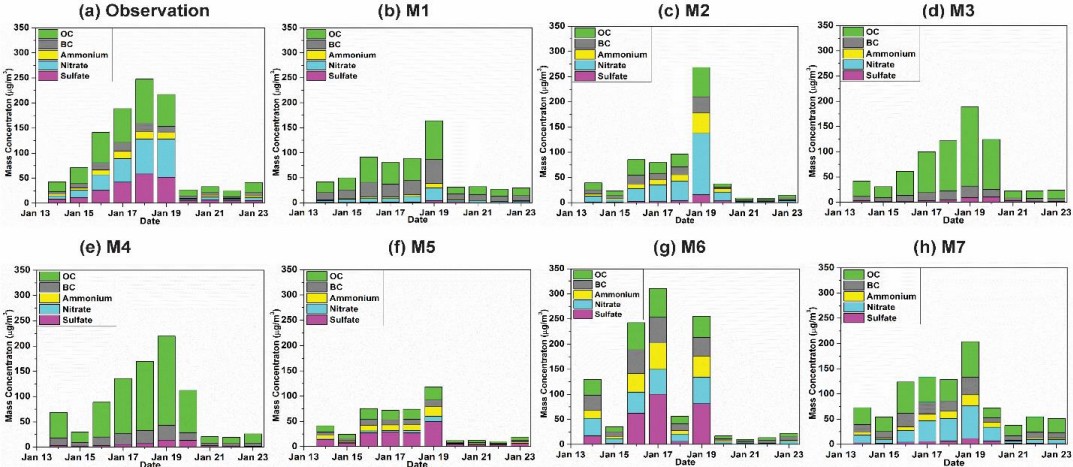

Figure 10. Observed and simulated daily mean concentrations of major PM$_{2.5}$ chemical

components in the urban Beijing site



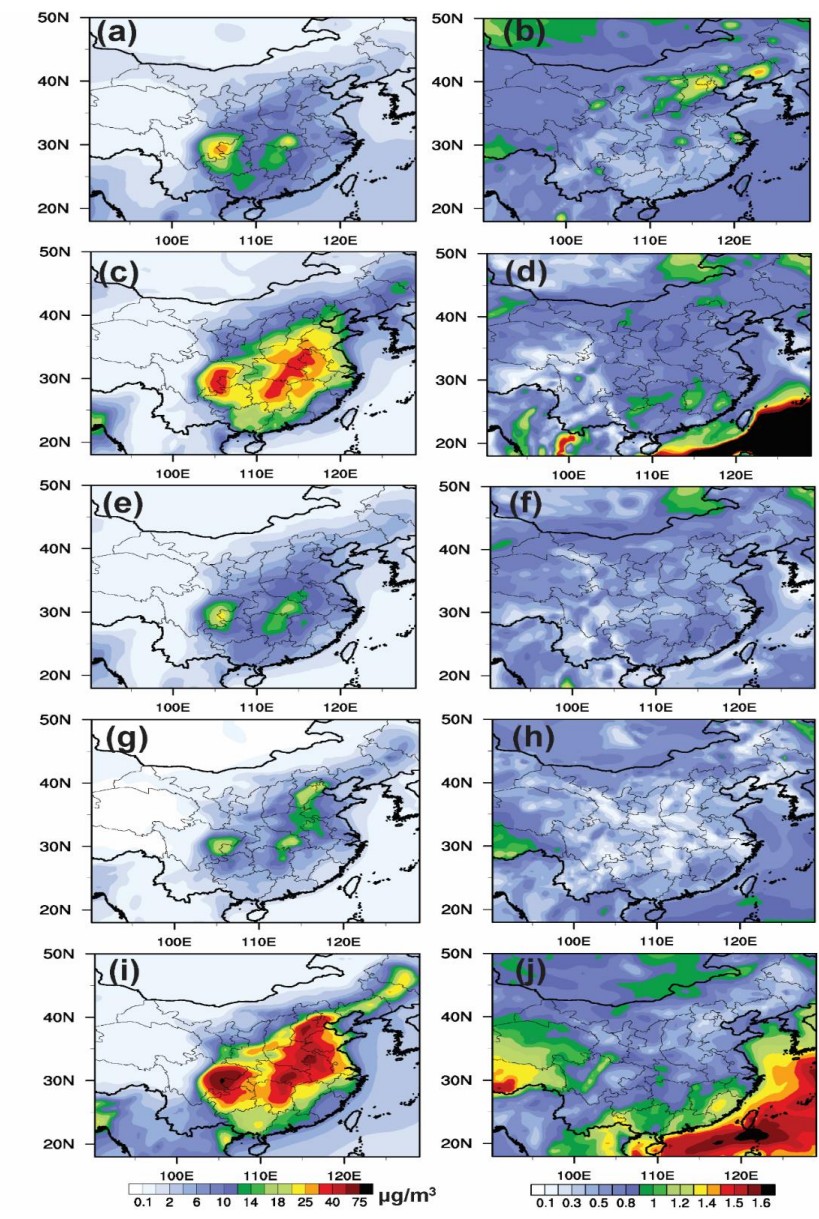

Figure 11. The ensemble mean monthly averaged near-surface distributions of PM$_{2.5}$

compositions for January 2010 (sulfate (a), nitrate (c), ammonium (e), BC (g), and OC (i)), along

with the spatial distribution of the coefficient of variation ((b), (d), (f), (h), and (j), standard

deviation divided by the average)


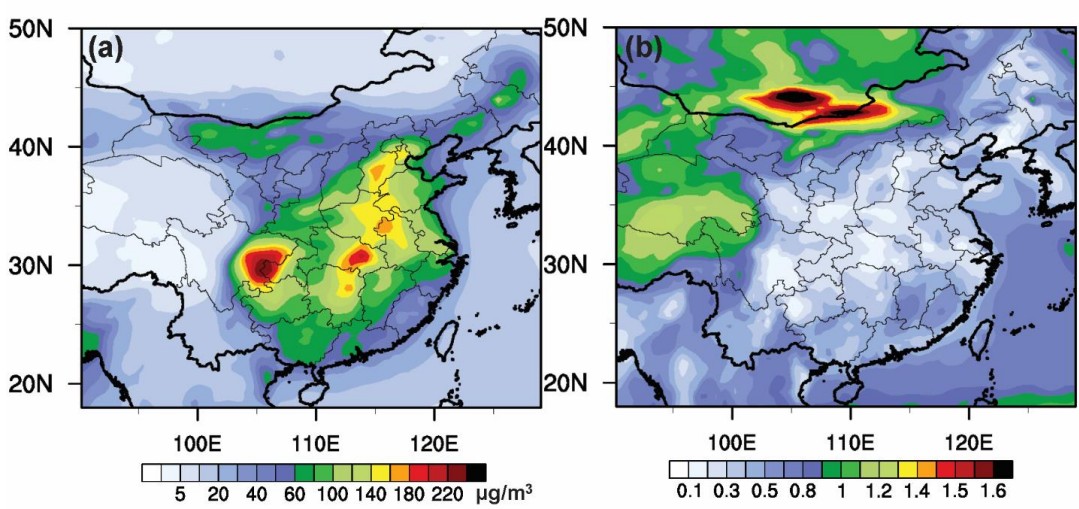

Figure 12. The ensemble mean monthly averaged near-surface distributions of PM$_{2.5}$ for January 2010 (a), along with the spatial distribution of the coefficient of variation (b, standard deviation divided by the average)

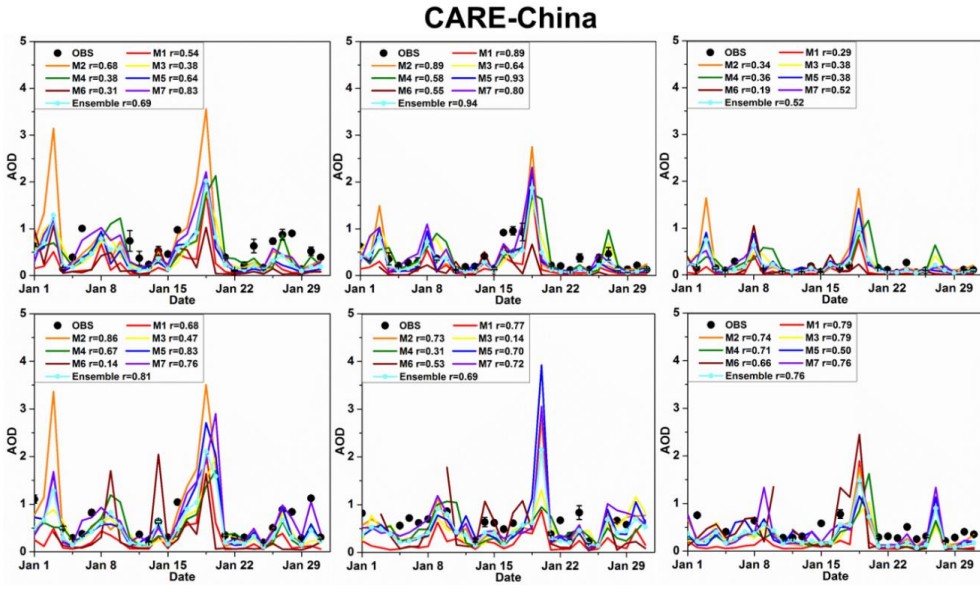




Figure 13. Comparisons between simulated and observed AOD at the CARE-China sites

(Baoding, Beijing City, Beijing Forest, Cangzhou, Jiaozhou, Shenyang,)

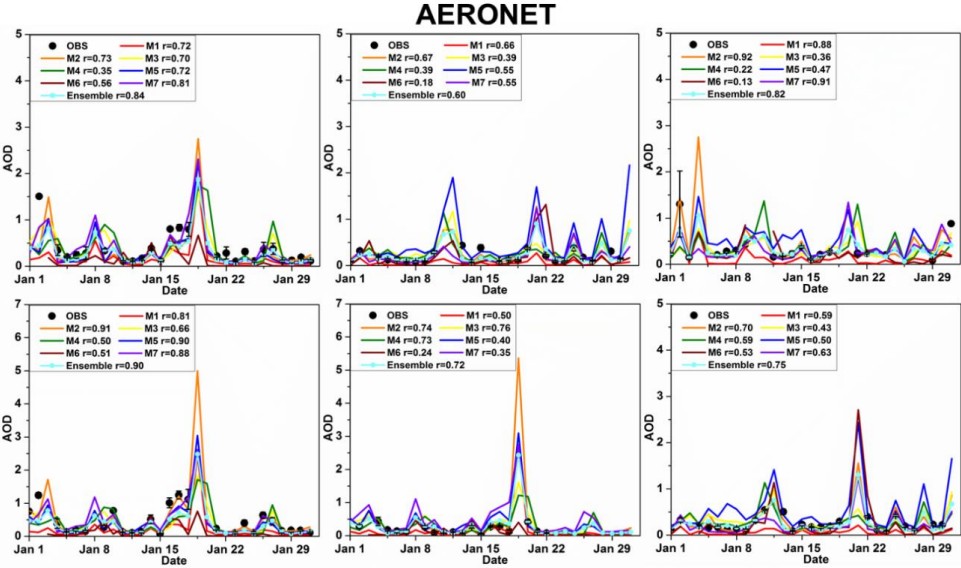

Figure 14. Comparisons between simulated and observed AOD at the AERONET sites (Beijing,

Shirahama, GIST, Xianghe, Xinglong, Osaka)