# Peer review of "Air Quality and Climate Change, Topic of the Model Inter-Comparison"

_Atmospheric Chemistry and Physics, 2017_

## Referee Comment (RC1) · Anonymous Referee #1 · 2 Oct 2017

This paper summarizes the overall performance of several fully-coupled air quality models that participated in the MICS-Asia III intermodal comparison study. It is the first part of a multi-part study. While the paper is well organized and the discussion is straight-forward, there are numerous places where the grammar needs to be fixed. I have tried to make some suggestions in my specific comments; however, the authors should not assume I have found all the problems. There are some aspects of the manuscript that are not explained well, such as the rationale for this paper in relation to future parts and the rationale of the intercomparison framework. By the time I finished

the paper, I feel like I did not learn anything substantially new; therefore, the authors have not adequately highlighted the new results of this study.

Major Comments:

1) In the introduction, the authors talk about Topic 3 of MICS-Asia phase 3 which is the subject of this paper. At the end of the paper I felt like I did not get any information regarding the feedbacks. Perhaps the paper title is implying that those details will be included in subsequent parts. It would be useful at the end of the introduction to have a clear understanding of what the objectives are for this paper, versus subsequent parts that will appear.

2) The authors do speculate why there are differences among the models; however, the paper would be improved significantly if they went into more detail into a few instances to find more concrete reasons for the differences. This might require more analysis of the results. But as the paper stands, it does not shed any new light on why the air quality models could differ. In the conclusion, the authors state that the paper provides "some directions of future model developments", but I see no evidence of that in the paper. The authors could also do a better job at citing papers that examine processes that might be missing or poorly represented in the air quality models.

3) The purpose of the MICS-Asia phase 3 is to look at feedback effects. I gather that is not that subject of this paper, and this paper is showing the initial evaluation of the aerosol simulations that will be important when looking at feedback effects later. The authors go into some detail on evaluating aerosol composition, but do not say anything about size distribution. Size distribution will be just as important for optical and CCN properties. I suggest adding a section that compares the predicted size distribution in some manner. I assume there are some sort of size distribution measurements that could be utilized. If not, it would still be useful in the context of the subsequent papers.

Specific Comments:

[Figure]

Line 43: Change "resolutions" to "resolution".

Line 56: I would change "are consistent" to "are similar". "consistent" can imply that the model results are good, but they could all consistently disagree with data. Change "haze event" to "haze events".

Line 58: "some brief senses" is an awkward phrase and should be replaced.

The abstract could be shortened so that it contains only the most important findings. For example, the sentences in lines 44 – 48 could be removed. The whole abstract is rather weak.

Line 66: Change "but primarily in Asia" to "but most deaths occur primarily in Asia."

Line 74: I would not use a semi-colon here and just have two sentences, although the second would need to be rephrased slightly.

Lines 134-139. There appears to be no underlying motivation for how the air quality models are compared. The only constraint on the models was the use of the same emissions inventories and they had to provide a set of variables. To better isolate the differences among the models, it would have been useful to have similar domains, grid spacings, and boundary conditions. I understand it would make the setting up the models a bit more difficult, but it would significantly reduce the differences arising from boundary conditions and spatial resolution. There are already many differences associated with the internal treatments of meteorology, chemistry, and aerosols. What I am looking for here is some further explanation as to why MICS-Asia organizers found the current configuration sufficient.

Line 153: The Grell reference is correct, but it only describes the initial model which did not discuss any of the feedback processes – which seems to be the focus of MICS-Asia Phase III. Those feedbacks were first implemented in Fast et al. (2006) and revised in subsequent manuscripts.

Line 182: For VBS, need to cite Adhamov et al. (2012) in which it was developed and

described.

Lines 190-191: The sentence regarding SOA is not correct and misleading. A VBS SOA treatment has been available in the public version of WRF-Chem for several years. What the authors mean to say is that the version of MOSAIC used in this study includes no SOA. The correct language here should imply that the users have chosen not to include SOA.

Section 2.1, in general. The description of the models is uneven. Some sections to into some details about the aerosol model, such as noting the geometric means of the modes (e.g. M5) but not going into the same detail as another model (e.g. M1). For one model the details of how optical properties and hygroscopicity are discussed, but then another model the same level of detail is not discussed. The authors need to revise this section to have the appropriate level of detail for all models.

Lines 232 – 239: The text discusses differences in the physics configurations as it should, but I assume the other models have physics differences too. Why not state that?

Line 286: What does regridding mean? To handle the emissions inventories properly, the emissions need to be reapportioned so that mass is neither gained or lost. Regridding implies interpolation, that to me indicates a poor method of handling the emissions from one domain to another.

Line 409: Change "are frequently happening" to "frequently happen".

Section 4, in general: this study relies on comparing model output to relatively few (at least for the PM data) point measurements. However, some discussion is needed to put the proper context of this type of comparison since the grid size differs among the models so there are issues of representativeness that must be considered.

Line 442: Turbulent mixing is missing from this description, which is not the same as transport.

[Figure]

Lines 469-483: What is missing from this discussion is how clouds affect the prediction of downward shortwave radiation. I assume that the clouds are the main factors controlling clouds, but there is no mention of this. Would be useful to include what the clear-sky values are in Figure 5.

Line 493: Change "larger near surface" to "larger near the surface".

Line 498: Awkward sentence – need to revise.

Line 513-514: Change to "All models produce similar CO predictions" based on how I understand this sentence.

Line 522: It is rather surprising that the models produce better PM than ozone. Usually it is the other way around.

Line 548: I doubt that sea-salt emissions are responsible for differences in PM10.

Line 569-580: Chen et al., ACP (2016) is just but one paper that describes possible missing reactions associated with sulfate. It would be useful for the authors to delve a bit deeper into the literature to find such issues associated with models. Most community models are inherently dated and do not necessarily have the most up-to-date chemistry treatments since it takes time for new research findings to make there was in to those community models.

Line 593-594: There are probably other reasons as well for errors in nitrate predictions.

Line 607: The authors list deposition, but this usually means dry deposition. What about wet scavenging? Same comment applies to line 610.

Line 607: Change predicted BC to "predicted BC at the surface"

Line 609: I think the authors mean horizontal grid resolution and not "horizontal grid interpolation." I have no idea what the latter means in this context. Please be more specific.

[Figure]

Line 612: Since POC is about the same from the models, then BC should be as well. So it is a bit of a mystery why BC from M2 and M7 are higher than the other models.

Line 621: Find a reference for this comment – there are lots of papers to cite here.

Line 638: This implies the model is missing a feedback, and I thought this study was about in the inclusion of feedbacks (see line 106 on topic 3).

Line 642: "dust deflation" is an odd phrase. What is that?

Lines 650-652: These sentences are poorly written. Suggest changing to "Only the sulfate predictions from M5 are close to the observed values. Sulfate is much lower than observed for all other models, except M6 which is too high. M2 and My predict reasonable nitrate concentrations. M3 and M4 overpredict OC during the haze period, but other models underpredict OC concentrations."

Line 677: What about clouds? How often were AOD retrievals not possible due to cloudy conditions?

Line 680: The figure captions should also state that the AOD is a daily (daytime) value.

Line 686: Change "it's" to "it is".

Line 718: Change "shows overprediction" to "shows an overprediction".

Line 724: Change "lower RH simulation" to "lower simulated RH".

Line 726: Change "OC concentration" to "OC concentrations".

Line 736-737: The authors have not shown this. It is very likely the size distribution and mixing state is treated differently. In this sense, the explanation provided previously in the paragraph is incomplete. I doubt one can really attribute the difference in AOD without a more rigorous analysis than the simple explanations presented here. At best, they are showing the range of AOD associated with all the differences among the models.

Lines 773-775: I don't see how interpolation of emissions to the grid should lead to model uncertainties. Of course, there could be errors introduced to reapportion emissions from one grid to another. But these would only be large if the mathematical method of reapportionment is poorly treated. There are ways to ensure that such uncertainties are small.

Line 776: "Manifold" is a strange word to use in this context.

Line 783: And what are those improvements? I would that that such an intercomparison study such as this would provide more concrete recommendations.

Line 788- 792: This sort of general conclusion about model inter-comparisons studies is rather tired. It could have been written based on results already in the literature and by speculation, without even conducting this inter-comparison study. I wish he authors could be more specific here regarding the findings specific to this study. The authors have not investigated all these possibilities and only barely scratched the surface at isolating and ascribing the uncertainties to specific processes.

Conclusion: It would be useful to have some text that looks forward to the next part of the paper. Will the authors be looking at evaluating the feedbacks, which were not examined in this study? Another reason I am looking for some more concrete explanations in this paper for why the models differ is that the situation will only become more complicated when examining feedback effects.

---

## Referee Comment (RC2) · Anonymous Referee #2 · 3 Nov 2017

The paper describes the setup of the MICS Asia Phase III model experiment. The results of seven simulations with online coupled meteorology-atmospheric chemistry models are shown and compared against observational data. The evaluation of on-line coupled air quality models and the outcome of the MICS-ASIA III model inter-comparison exercise are certainly worth to be published. However, the quality of the paper must be improved significantly before it can be published in ACP.

In the first instance, the 'Results and discussions' section must be enhanced considerably. Attempts should be made to explain the reasons for the observed differences

among the models. In particular, a more in depth discussion is necessary for those model results which look like outliers (for example solar radiation and ozone in Figure 5 and 6).

Although the paper is overall well organized, it was nevertheless a though read for me. It is sometimes difficult to keep track of the different models and their respective se-tups. Repetition of the model name along with the label once a while could improve this situation with only little effort. Enhancing the figure captions would also help. Fi-nally, several sentences are quite convoluted and hard to understand. Splitting long sentences may improve the readability.

For these reasons and further reasons mentioned below, major revisions are required. The specific comments include some suggestions how to improve the paper.

**Specific comments (also including minor points):**

Abstract: In its current form the abstract raises the expectation that aerosol meteorol-ogy interactions will also be a topic of this paper. The reader may also expect that both episodes will be evaluated in the paper. Generally, the first part of the abstract is raising expectations, which are not fulfilled by the paper. Nevertheless, it is also OK to restrict the paper to the evaluation of first episode. However, the abstract must be reworded in this case in order to avoid raising expectations which are not met later. It is not clear why the model evaluation is restricted to the first episode of the model inter-comparison and why the second episode is not considered.

Introduction: The authors should also consult publications describing related work, e.g. the model evaluation papers related to HTAP as well as AQMEII Phase 2 and Phase 3.

Line 106: 'Various multi-scale models . . .' Are there more than the five model and seven simulations described here?

Lines 114 –119: The references should not be restricted to Chinese authors.

Line 125: The paper claims to be 'serving as the main repository of the information linked to Topic 3 simulations and comparisons.' To achieve this aim, more details must be added to the descriptions of those models where die description is quite. Please add also some paragraphs in section 2.1 (similar to section 2.5) in order to make this section better readable.

Table 1: The numbers attached to 'WRF-Chem' and 'NU-WRF' are unnecessary and confusing. Therefore, they must be removed. On the other hand, the model version is an important information which must be added wherever applicable.
The contents of the table are not precise: For example, for M1 'RACM' must be replaced by 'RACM-ESRL' and 'MADE' by 'MADE/VBS'. What climatological data are applied as boundary conditions for M6?
Please consider to add also information about the details of the radiation calculation. As shown by Curci et al. (2015, Atmospheric Environment, Vol. 115) the inherent assumption have a strong influence on the calculated AODs.

Table S1: Please check also whether this table needs to be more specific (similar to Table 1). Please add also information about the two models which are missing in the table.
Please consider also moving this table into the main part of the paper.

Line 338: Please mention the source of the climatological data her or in Table 1.

Line 281: Please add the information which model uses the prescribed BVOC emissions and which one does the internal calculation (could also be added to Table 1).

Section 2.1: Please mention clearly in the text which aerosol meteorology interactions are switched on for each model.

Section 2.2: Please add some information on soil dust emissions (also to be included in Table 1).

Section 2.3: Please include some information about the meteorological boundary conditions.

Section 2.3: Please consider to add some information about the differences between the boundary values from the different data sources for selected variables (eventually to be included in the supplementary material). Looking into the boundary values may also help to understand differences between results of the different model runs.

Line 389: This sentence is confusing. According to Table S1 all simulations considered here are performed with aerosol meteorology interactions switched on. Were the simulations additionally performed also without aerosol meteorology interactions for the investigation of feedback effects? Anyway, this could be mentioned in the introduction as well as in the outlook, but not at this place.

Line 392 –393: Why are abbreviations given for temperature, humidity, and wind, and units for the shortwave radiation?

Section 3: Why is the year 2013 described here, if it not discussed in the rest of the paper? Figure 3 could eventually be moved to the supplementary material.

Section 4.1: Cloud optical depth and integrated liquid water are important and should also be discussed (even, if no observational data are available).

Line 427: This topic is not addressed in this paper.

Line 473: What is the reason for the bad performance of M6 and M7? Please check also cloud cover. And why is this huge difference in radiation between M6 and M7 not reflected in T2? Why does M7, which is also WRF, show such a large difference to M1 and M2?

Line 480: Why is this the case?

Section 4.2: Spatial distributions of the gas phase pollutants (similar to figures 8 and 9) could be shown also in the supplement.

Line 498: The 'only' is probably placed wrong here?

Line 504: Plese give some evidence for this.

Line 521: Please give more evidence for this statement. And what is the contribution of soil dust to PM10? What is the contribution of different of the dust emission parameterizations?

Line 524 – 526: What is the reason for the negative correlation? On an hourly basis, the diurnal course of ozone should reflect the course of the solar radiation. Or are the correlations just calculated from on daily values? Please clarify. How do the diurnal courses look like for the individual models? Could the parameterization of dry deposition or differences in the lateral boundary conditions or differences in biogenic VOC emissions explain the overestimation of simulated ozone for M3 and M4? It is surprising, that the overestimated ozone seems not to be related to solar radiation, so there are probably other reasons for this overestimation. How do the ozone profiles look like, can they contribute to an explanation?

Line 530: If the citation does not fit, it should be removed. Knote et al. (2015, Atmospheric Environment, Vol. 115) may probably suit better as this paper includes also a winter episode.

Sections 4.3 and 4.4: It would be nice if soils dust were also included in the discussion.

Lines 550 – 554: These general remarks are not necessary here.

Line 561: 'M5 and M6 shows . . .': Please observe the proper use of singular and plural (not only here but throughout the paper, e.g. line 582).

Lines 562 – 578: It is not clear in how far these statements apply to the models which are discussed here.

Line 568 – 572: This sentence is quite hard to understand. Please split it into two or more sentences. Please also split other lengthy convoluted sentences.

Line 593: Is this just a general remark or is there some evidence for this? Does M7

really include heterogeneous nitrate formation?

Line 608 and line 772: How can this be? Emissions were supposed to be the same for all models.

Line 609: Was the vertical distribution of the emissions not prescribed? If there are differences in the vertical distributions, they must be described.

Line 616: According to line 191, M2 does not include SOA formation . . .

Line 616 – 617: The statement about SORGAM does not fit here as M1 includes a VBS approach (Ahmadov, R., et al., 2012, J. Geophys. Res.).

Line 619: 'volatile' seems to be missing here

Lines 619 – 624: These lines include quite general statements. How are they related to the models discussed in this paper?

Lines 642 – 643: This should have been mentioned in section 2.2.

Lines 669 – 672: Please avoid this kind of redundancies (not only here).

Lines 677 – 680: What is the contribution of soil dust during these situations?

Line 688: Please split this sentence.

Line 690 – 691: This statement is true, but unnecessary. Lines 692 – 798: These are quite general statements, which must be related to the applied models.

Line 710: M2 does not have modes . . .

Line 712: What are the consequences of this? If necessary, this can be discussed based on the findings by Curci et al. (2015, Atmospheric Environment, Vol. 115).

**Figures and figure captions:**

Figure captions in general: More detailed descriptions must be given in the figure

captions (are daily values or hourly values shown, relevant area, etc., depending on what is shown in the figure).

Figure 1: It looks like the labels M3 and M4 in the legend of Figure 1 are mixed up (According to Table 1 and Figures 8 and 9 M4 is the small domain). Furthermore, using the same colors for the model domains as for the curves in Fig. 5 etc. would make the reading a bit easier.

Caption of Figure 1: Repeating the model names in the caption (for example 'M1: WRF-Chem, 45 km; M2 WRF-Chem, 50 km, M3 ...') would make the paper a bit 'reader-friendly'.

Caption of Figure 5: Please mention that this is a spatial average (for a, b and c: over which area) of daily values and explain error bars.

Figure 7: Please show also CO and PM2.5 although no measurements are available.

Figures 8 and 9: Please include also PM10.

Figures 8 and 9: The split into two figures appears quite arbitrary. Perhaps it would look better if the figures are organized differently: One figure with 7 rows (M1 – M7) and 3 columns (PM10, PM2.5, BC) and one figure with 7 rows (M1 – M7) and 4 columns ($SO_4^{2-}$, $NO_3^-$, $NH_4^+$, OC) – no obligation to do this, just a suggestion.

Spatial distributions of the gas phase pollutants (similar to Figures 8 and 9) would be nice in the supplement.

Figure S4: This figure seems to be contorted. Please improve the quality. Why are all the lines within the single 'height groups' (e.g. at '1 km') at different heights? Please explain in the figure caption. Also: explain what is shown here (daily values, hourly values, ...?)

---

## Author Comment (AC1) · 23 Dec 2017

We would like to express our sincere thanks to the two reviewers for their careful reading and constructive suggestions, which have helped us improve the quality of this manuscript. We have addressed all their comments carefully and revised the manuscript accordingly. The detailed responses to their concerns and comments are presented as follows.

**Anonymous Referee #1**

This paper summarizes the overall performance of several fully-coupled air quality models that participated in the MICS-Asia III intermodal comparison study. It is the first part of a multi-part study. While the paper is well organized and the discussion is straight-forward, there are numerous places where the grammar needs to be fixed. I have tried to make some suggestions in my specific comments; however, the authors should not assume I have found all the problems. There are some aspects of the manuscript that are not explained well, such as the rationale for this paper in relation to future parts and the rationale of the intercomparison framework. By the time I finished the paper, I feel like I did not learn anything substantially new; therefore, the authors have not adequately highlighted the new results of this study.

Major Comments: 1) In the introduction, the authors talk about Topic 3 of MICS-Asia phase 3 which is the subject of this paper. At the end of the paper I felt like I did not get any information regarding the feedbacks. Perhaps the paper title is implying that those details will be included in subsequent parts. It would be useful at the end of the introduction to have a clear understanding of what the objectives are for this paper, versus subsequent parts that will appear.

Response: Thanks for this great suggestion. In the last paragraph of introduction, we claim that "This paper presents and overview of the MICS-ASIA III Topic 3, serving as the main repository of the information linked to Topic 3 simulations and comparisons". To make it clear, we added in the manuscript: "Specifically, this paper aims to archive the information of participating models, how the experiments, and results of model evaluation. The results of the MICS-Asia Topic 3 experiments looking at the direct and indirect 135 effects during heavy haze events will be published in a companion paper, part II."

2) The authors do speculate why there are differences among the models; however, the paper would be improved significantly if they went into more detail into a few instances to find more concrete reasons for the differences. This might require more analysis of the results. But as the

paper stands, it does not shed any new light on why the air quality models could differ. In the conclusion, the authors state that the paper provides "some directions of future model developments", but I see no evidence of that in the paper. The authors could also do a better job at citing papers that examine processes that might be missing or poorly represented in the air quality models.

Response: Thank you for pointing out. We added more explanations on why model differs in the revised manuscript. For example, the radiation differences in M6 is due to the use of different meteorological boundary conditions, and the averaged values were taken from 3hourly outputs. More importantly, high liquid water path in M6 lead to lower radiation from M6 in north China.

We have cited more papers on the reasons for underestimation of sulfate and SOA, and clearly claim the future directions for future model developments: "X. Huang et al. (2014) found including natural and anthropogenic mineral aerosols can enhance sulfate production through aqueous-phase oxidation of dissolved SO2 by O3, NO2, H2O2 and transition metal. Gao et al. (2016), Wang et al. (2014), and Zhang et al. (2015) also emphasized the importance of multiphase oxidation in winter sulfate production. However, these processes are currently not incorporated in the participating models for this study, which might be responsible for the apparent under-predictions of sulfate concentration";

"An et al. (2013) incorporated photoexcited nitrogen dioxide molecules, heterogeneous reactions on aerosol surfaces, and direct nitrous acid (HONO) emissions into the WRF-Chem model and found these additional HONO sources can improve simulations of HONO and nitrate in north China. M7 also predict high nitrate concentrations (N2O5 and NO2 gases react with liquid water, Zheng et al., 2015), and the predicted lower nitrate concentrations from other models are probably due to missing aqueous phase and heterogeneous chemistry, or the implementations of different gas phase oxidation in these models. Many studies have been conducted regarding sulfate formation issues. Nitrate also account for a large mass fraction in PM2.5 during winter haze events in north China, yet less attention was attracted to fully understand its formation. It is worth furtherly digging into the details about how different processes contribute to high nitrate concentrations in future studies";

"R. Huang et al. (2014) also suggested that low temperature does not significantly reduce SOA

formation rates of biomass burning emissions. Most models over-simplified SOA formation"; "Zhao et al. (2015) comprehensively assessed the effect of organic aerosol aging and intermediate-volatile emissions on OA formation and confirmed their significant roles. All these results suggest more complicated SOA scheme are needed to improve organic aerosol simulations during haze events".

3) The purpose of the MICS-Asia phase 3 is to look at feedback effects. I gather that is not that subject of this paper, and this paper is showing the initial evaluation of the aerosol simulations that will be important when looking at feedback effects later. The authors go into some detail on evaluating aerosol composition, but do not say anything about size distribution. Size distribution will be just as important for optical and CCN properties. I suggest adding a section that compares the predicted size distribution in some manner. I assume there are some sort of size distribution measurements that could be utilized. If not, it would still be useful in the context of the subsequent papers.

Response: Thanks for this good suggestion. We are aware that size distribution is crutial to optical properties and CCN formation, and thus direct and indirect effects. The participating models use mode and sectional approaches with different degree of complexity, we present the basic information on size distribution in each model, such as geometric mean radius and standard deviation and the number and range of size bin, and briefly mention about its potential influence on AOD simulation. We know this is not enough and we will investigate this issue in more detail in the companion paper (part II) by conducting additional sensitivity simulations with different size distribution while keeping other conditions the same. For this paper, we add more information on aerosol size distributions in the models in Table 1. Except M2 using 8-bin MOSAIC sectional approach, the other models use mode and bulk approach, with similar mean radius and standard deviation for anthropogenic aerosols or no size information. Unfortunately, we don't measurement of size distribution for comparison.

**Specific Comments:**

Line 43: Change "resolutions" to "resolution".

Response: We have changed.

Line 56: I would change "are consistent" to "are similar". "consistent" can imply that the model results are good, but they could all consistently disagree with data. Change "haze event" to "haze events".

Response: We have changed.

Line 58: "some brief senses" is an awkward phrase and should be replaced. The abstract could be shortened so that it contains only the most important findings. For example, the sentences in lines 44 – 48 could be removed. The whole abstract is rather weak.

Response: We have changed "provide some brief sense of" to "present"; we remove lines 44-48. This abstract is mainly used to introduce Topic 3 and how simulations were done and analyzed. Major findings out of Topic 3 will be published in the following companion papers.

Line 66: Change "but primarily in Asia" to "but most deaths occur primarily in Asia."

Response: We have made this change.

Line 74: I would not use a semi-colon here and just have two sentences, although the second would need to be rephrased slightly.

Response: We have made this change.

Lines 134-139. There appears to be no underlying motivation for how the air quality models are compared. The only constraint on the models was the use of the same emissions inventories and they had to provide a set of variables. To better isolate the differences among the models, it would have been useful to have similar domains, grid spacings, and boundary conditions. I understand it would make the setting up the models a bit more difficult, but it would significantly reduce the differences arising from boundary conditions and spatial resolution. There are already many differences associated with the internal treatments of meteorology, chemistry, and aerosols. What I am looking for here is some further explanation as to why MICS-Asia organizers found the current configuration sufficient.

Response: We tried our best to constrain the differences, by requesting the modeling groups to use the same emission inventories, domains, grid size and boundary conditions, but some models has been constructed and tested on their own configurations, such as M5 and M6, and all the modeling groups use a similar resolution ~50km (Table 1) except M4 (15km). M4 actually covers a nested domain of M3. We include this simulation to further look at the impacts of grid spacing on aerosol feedbacks, which will be documented in the companion paper.

Models are quite different from each other, and it is difficult to keep all the inputs the same. Even using the same boundary conditions, mapping species from global chemistry simulations to different chemical mechanism each model uses can also lead to differences. We describe what we provided in the methodology section was intended to reduce the differences in inputs. Results from these model configurations have been published to report aerosol feedbacks and aerosol direct and indirect effects in Asia. The underlying motivation is to look at how current reported values differ, and provide possible range of these values. Given this motivation and our efforts on constraints, we think our current configuration could be sufficient although not ideal.

Line 153: The Grell reference is correct, but it only describes the initial model which did not discuss any of the feedback processes – which seems to be the focus of MICS-Asia Phase III. Those feedbacks were first implemented in Fast et al. (2006) and revised in subsequent manuscripts.

Response: We have added Fast et al. (2006).

Line 182: For VBS, need to cite Adhamov et al. (2012) in which it was developed and described.

Response: We confirmed that M1 did not use VBS, so we deleted description here.

Lines 190-191: The sentence regarding SOA is not correct and misleading. A VBS SOA treatment has been available in the public version of WRF-Chem for several years. What the authors mean to say is that the version of MOSAIC used in this study includes no SOA. The correct language here should imply that the users have chosen not to include SOA.

Response: We changed it to "The MOSAIC version used in M2".

Section 2.1, in general. The description of the models is uneven. Some sections to into some details about the aerosol model, such as noting the geometric means of the modes (e.g. M5) but not going into the same detail as another model (e.g. M1). For one model the details of how optical properties and hygroscopicity are discussed, but then another model the same level of detail is not discussed. The authors need to revise this section to have the appropriate level of detail for all models.

Response: Thanks for these good suggestion. We reorganize the whole section to make it easier to read and keep details for all models even.

Lines 232 – 239: The text discusses differences in the physics configurations as it should, but I assume the other models have physics differences too. Why not state that?

Response: Thanks for these good suggestion. We have added other physical configurations in Table 1 and in the texts.

Line 286: What does regridding mean? To handle the emissions inventories properly, the emissions need to be reapportioned so that mass is neither gained or lost. Regridding implies interpolation, that to me indicates a poor method of handling the emissions from one domain to another.

Response: Yes, the emissions were handled using mass conservative method to make sure that mass is neither gained nor lost, not simple interpolation.

Line 409: Change "are frequently happening" to "frequently happen".

Response: We have changed.

Section 4, in general: this study relies on comparing model output to relatively few (at least for the PM data) point measurements. However, some discussion is needed to put the proper context of this type of comparison since the grid size differs among the models so there are issues of representativeness that must be considered.

Response: This is a good point. We required the modeling groups use similar model resolutions. For most models, ~50km resolution is used. Except model evaluation, the model resolution would also affect simulations of meteorological fields and then chemical fields. We are preparing a manuscript discussing the influences of model resolution be keeping all other factors the same, which will be published in this topic series. Based on this, we think the comparison representativeness would be close for participating models. For model observation comparison, this is a common problem, given limited available observations in this study. In our previous experiences, the comparisons are done mostly in urban regions, and measurements during haze events show high homogeneity when we compared concentrations from different city sites in previous study. Besides, the in situ measurements at city sites exhibit a similar variation trend during haze events, reflecting haze pollution at regional scale rather than local scale, thus the concentration difference resulting from grid size would reduce.

Line 442: Turbulent mixing is missing from this description, which is not the same as transport.

Response: We have added turbulent mixing.

Lines 469-483: What is missing from this discussion is how clouds affect the prediction of downward shortwave radiation. I assume that the clouds are the main factors controlling clouds, but there is no mention of this. Would be useful to include what the clear-sky values are in Figure 5.

Response: Thanks for this great point. Since clear sky radiation is not submitted by modeling groups and it is not a default model outputs, it might not be appropriate to show it in Figure 5. However, we do think it is important to include the discussion here because we agree with you that it is one of the main factors controlling radiation. We add a figure to show the reduction ratio of downwards shortwave radiation due to clouds (SW-SWclearsky)/ SWclearsky in the SI, and discussion on clouds: "Clouds are also important to alter radiation. To exclude its impacts on the radiation shown here, we calculated the reduction ratio of radiation due to clouds. During the severe haze period (16-19 January 2010), the averaged reduction fraction is 5.9% in north China and 4.2% in south China, suggesting the relatively lower radiation during this period shown in Figure 5(d) is mainly caused by aerosols, while the lowest radiation on 20 January was caused by clouds (Figure 5(d))."

[Figure]

Line 493: Change "larger near surface" to "larger near the surface".

Response: We have changed.

Line 498: Awkward sentence – need to revise.

Response: We removed "only".

Line 513-514: Change to "All models produce similar CO predictions" based on how I understand this sentence.

Response: We have changed.

Line 522: It is rather surprising that the models produce better PM than ozone. Usually it is the other way around.

Response: This is probably because previous good ozone simulations mostly occur in summer. The study period is frequently affected by heavy haze in winter, when photochemistry is weaker and other chemical processes maybe more complicated. Another reason is that primary PM (including POA, BC etc) is of similar magnitude to secondary aerosol in winter haze period, which means meteorological and chemical processes are equally important.

Line 548: I doubt that sea-salt emissions are responsible for differences in PM10.

Response: We recheck the model and possible reason for the differences in PM10. Fig 7d shows that M5 largely overpredict PM10 concentration during 1-8 January at Rishiri, where is less affected by continental aerosols, overestimation of sea salt emission under certain meteorological conditions (such as large wind) is responsible for such positive biases, whereas in the rest of January, M5 perform relatively better and is generally consistent with most of the models.

Line 569-580: Chen et al., ACP (2016) is just but one paper that describes possible missing reactions associated with sulfate. It would be useful for the authors to delve a bit deeper into the literature to find such issues associated with models. Most community models are inherently dated and do not necessarily have the most up-to-date chemistry treatments since it takes time for new research findings to make there was in to those community models.

Response: We have added several papers describing the model problems and how missing mechanism can explain the underestimation of sulfate and other model problems. All the participating models have not implemented such mechanism, which is an important source for the underprediction of sulfate in this study.

Line 593-594: There are probably other reasons as well for errors in nitrate predictions.

Response: Thanks for mentioning this. We added the spatial plots of NOx in the SI. It is shown that NOx from M5 is the highest, yet the nitrate produced from M5 is lower than other models except M1, which suggest there could be some missing nitrate formation pathways or stronger deposition of nitrate and its precursors in M5. Other than the mentioned aqueous phase and heterogeneous chemistry, the implementation of gas phase chemistry in different models might also played a role here. We added this to the revised sentence: ", or the implementations of different gas phase oxidation in these models. Many studies were conducted for sulfate formation issues. Nitrate also account for a large mass fraction in PM2.5 during winter haze

events in north China, yet less attention was attracted to fully understand its formation. It is worth furtherly digging into the details about how different processes contribute to high nitrate concentrations in future studies."

Line 607: The authors list deposition, but this usually means dry deposition. What about wet scavenging? Same comment applies to line 610.

Response: Thanks for this point. Here we meant both dry and wet deposition. To make it clearer, we added "(dry deposition and wet scavenging)". We added a figure in the SI to check how often were clouds present during this month. There were clouds only in a few days, but they can be very important for wet scavenging for hydrophilic aerosols. Thus, we added wet scavenging in Line 607 and 610.

[Figure]

Line 607: Change predicted BC to "predicted BC at the surface"

Response: We have changed.

Line 609: I think the authors mean horizontal grid resolution and not "horizontal grid interpolation." I have no idea what the latter means in this context. Please be more specific.

Response: We have changed.

Line 612: Since POC is about the same from the models, then BC should be as well. So it is a bit of a mystery why BC from M2 and M7 are higher than the other models.

Response: It depends on the models treat deposition and aging processes. For example, in the GOCART aerosol model (M3 and M4), 80% of BC are assumed to be hydrophobic and then undergo aging to become hydrophilic in an e-folding time of 1.2 days. Hydrophilic aerosols will go through wet deposition. But in other models like M2 and M7, BC is assumed to be hydrophobic, thus the wet removal is less actived. We added these explanations in the revised manuscript.

Line 621: Find a reference for this comment – there are lots of papers to cite here.

Response: Thanks. We added the following citation.

Heald, C. L., Henze, D. K., Horowitz, L. W., Feddema, J., Lamarque, J.-F., Guenther, A., Hess, P. G., Vitt, F., Seinfeld, J. H., Goldstein, A. H., and Fung, I., 2008. Predicted change in global secondary organic aerosol concentrations in response to future climate, emissions, and land use change, J. Geophys. Res., 113, D05211, doi:10.1029/2007JD009092.

Line 638: This implies the model is missing a feedback, and I thought this study was about in the inclusion of feedbacks (see line 106 on topic 3).

Response: Yes, we are trying to say less VOCs should be able to convert to SOA under hazy conditions, but the model still used 10% yield, which could overestimates SOA.

Line 642: "dust deflation" is an odd phrase. What is that?

Response: We change to "wind-blown dust"

Lines 650-652: These sentences are poorly written. Suggest changing to "Only the sulfate predictions from M5 are close to the observed values. Sulfate is much lower than observed for all other models, except M6 which is too high. M2 and M7 predict reasonable nitrate concentrations. M3 and M4 overpredict OC during the haze period, but other models underpredict OC concentrations."

Response: We have changed following your suggestions.

Line 677: What about clouds? How often were AOD retrievals not possible due to cloudy conditions?

Response: Yes, it is possible. But in winter, there is relatively less cloud amount, so cloud is not a serious problem to AOD retrieval. For example, in Figure 14 and SI, about three days within a month (Jan. 2 Jan. 8 Jan 20) are due to clouds. We have added the influences of clouds in the text: "under serious pollution and cloudy conditions". Thanks for pointing out.

[Figure]

Line 680: The figure captions should also state that the AOD is a daily (daytime) value.

Response: We have changed following your suggestions: "daily (daytime) mean"

Line 686: Change "it's" to "it is".

Response: We have changed following your suggestions.

Line 718: Change "shows overprediction" to "shows an overprediction".

Line 724: Change "lower RH simulation" to "lower simulated RH".

Response: We have changed following your suggestions.

Line 726: Change "OC concentration" to "OC concentrations".

Response: We have changed following your suggestions.

Line 736-737: The authors have not shown this. It is very likely the size distribution and mixing state is treated differently. In this sense, the explanation provided previously in the paragraph is incomplete. I doubt one can really attribute the difference in AOD without a more rigorous analysis than the simple explanations presented here. At best, they are showing the range of AOD associated with all the differences among the models.

Response: Thank you for this question and suggestion. We added the mixing state of each model in Table 1. Only M6 used external mixing. Curci et al. (2015) discussed the impacts of mixing state on simulated AOD and found that external mixing state assumption significantly increase simulated AOD. M6 used external mixing but shows a relative lower AOD because of ignorance of other aerosol species like dust, sea-salt, etc.. Other models used internal mixing for major aerosol compositions. The size distribution treated in the models except M2 (sectional approach) and GOCART bulk approach is described by a lognormal distribution with similar geometric mean radius and standard deviation for different modes or species, such as

0.07 μm for inorganic aerosols, 0.01μm and 0.02 μm for BC and OC, but different bins for dust and sea salt (their concentrations are low in the north China Plain in winter). Thus we believe the differences in AOD shown here is mostly due to differences in simulated compositions. The size distribution in this sentence means the used modal size treatments in M1, M5, M6 and M7. To avoid misunderstanding, we change it to "lognormal treatments" and add the conclusion from Curci et al., (2015) about the impacts of mixing state.

Curci, G., et al. "Uncertainties of simulated aerosol optical properties induced by assumptions on aerosol physical and chemical properties: An AQMEII-2 perspective." Atmospheric Environment 115 (2015): 541-552.

Lines 773-775: I don't see how interpolation of emissions to the grid should lead to model uncertainties. Of course, there could be errors introduced to reapportion emissions from one grid to another. But these would only be large if the mathematical method of reapportionment is poorly treated. There are ways to ensure that such uncertainties are small.

Response: Thanks. We changed this sentence to "which might be caused by the treatment of aging and deposition (dry deposition and wet scavenging) processes."

Line 776: "Manifold" is a strange word to use in this context.

Response: We have changed to "various".

Line 783: And what are those improvements? I would that that such an intercomparison study such as this would provide more concrete recommendations.

Response: The above results provide some directions for future model development, such as new heterogeneous or aqueous pathways for sulfate and nitrate formation under hazy condition, SOA chemical mechanism with new VOC precursors, yield data and approaches, and the dependence of aerosol optical properties on size distribution and mixing state. We have added this in the revised manuscript.

Line 788- 792: This sort of general conclusion about model inter-comparisons studies is rather tired. It could have been written based on results already in the literature and by speculation, without even conducting this inter-comparison study. I wish he authors could be more specific here regarding the findings specific to this study. The authors have not investigated all these possibilities and only barely scratched the surface at isolating and ascribing the uncertainties to specific processes.

Response: This paper presents model evaluation and intercomparison for meteorological fields, aerosol concentrations and optical properties with a series of observations over East Asia. While these models exhibit a generally good performance for PM2.5 concentration, the simulations of chemical compositions differ largely, which lead to the large differences in optical properties, such as AOD, and this would further affect direct and indirect aerosol effects, and consequently radiation and cloud in this region. We will compare and examine how the different AOD levels and aerosol properties affect radiation and cloud and explore the strength and affecting factors of aerosol-radiation-weather interactions in the companion paper part II.

Conclusion: It would be useful to have some text that looks forward to the next part of the paper. Will the authors be looking at evaluating the feedbacks, which were not examined in this study? Another reason I am looking for some more concrete explanations in this paper for why the models differ is that the situation will only become more complicated when examining feedback effects.

Responses: Thank you for the good suggestion. We briefly describe the objective of the next part of the paper in the conclusion: "This paper focused on the evaluation of the predictions of meteorological parameters and the predictions of aerosol mass, composition and optical depth. These factors play important roles in feedbacks impacting weather and climate through radiative and microphysical processes."

**Referee #2**

The paper describes the setup of the MICS Asia Phase III model experiment. The results of seven simulations with online coupled meteorology-atmospheric chemistry models are shown and compared against observational data. The evaluation of online coupled air quality models and the outcome of the MICS-ASIA III model inter-comparison exercise are certainly worth to be published. However, the quality of the paper must be improved significantly before it can be published in ACP. In the first instance, the 'Results and discussions' section must be enhanced considerably. Attempts should be made to explain the reasons for the observed differences among the models. In particular, a more in depth discussion is necessary for those model results which look like outliers (for example solar radiation and ozone in Figure 5 and 6). Although the paper is overall well organized, it was nevertheless a though read for me.

It is sometimes difficult to keep track of the different models and their respective setups. Repetition of the model name along with the label once a while could improve this situation with only little effort. Enhancing the figure captions would also help. Finally, several sentences are quite convoluted and hard to understand. Splitting long sentences may improve the readability.

For these reasons and further reasons mentioned below, major revisions are required.

The specific comments include some suggestions how to improve the paper.

Response: We appreciate the detailed and constructive comments from referee #2 to help improve our manuscript. We have added more explanations for the shown model differences in the revised version. We also add model names in figure captions, rewrite some sentences, and split long sentences to improve the readability. The detailed point-by-point responses and changes are listed below.

**Specific comments (also including minor points):**

Abstract: In its current form the abstract raises the expectation that aerosol meteorology interactions will also be a topic of this paper. The reader may also expect that both episodes will be evaluated in the paper. Generally, the first part of the abstract is raising expectations, which are not fulfilled by the paper. Nevertheless, it is also OK to restrict the paper to the evaluation of first episode. However, the abstract must be reworded in this case in order to

avoid raising expectations which are not met later. It is not clear why the model evaluation is restricted to the first episode of the model inter-comparison and why the second episode is not considered.

Response: Thanks for pointing out. This manuscript is the overview paper for aerosol meteorology interactions topic, so we include it in the abstract to raise interests from audience in the companion papers. We deleted "Two winter months (January 2010 and January 2013) were selected as study periods, when severe haze occurred in North China." in the abstract to avoid the expectation. We restricted to the first case because all models submitted results for January 2010 and less models for January 2013. For other topics within MICS-ASIA III, only year 2010 was simulated. Besides, more measurements data are available for year 2010. We also found that we had enough materials (mostly project overview) to present for current manuscript, so decided to move the evaluations of the second case into part II manuscript. We have removed all descriptions for 2013 in the manuscript.

Introduction: The authors should also consult publications describing related work, e.g. the model evaluation papers related to HTAP as well as AQMEII Phase 2 and Phase 3.

Response: It is a great idea to include information from HTAP and AQMEII projects. In the introduction, we add "Other ongoing related modeling frameworks include the Task Force on Hemispheric Transport of Air Pollution (TF HTAP) and the Air Quality Model Evaluation International Initiative (AQMEII). The TF HTAP was initiated to improve knowledge of the intercontinental or hemispheric transport and formation of air pollution, and its impacts on climate, ecosystems and human health (Galmarini et al., 2017; Huang et al., 2017). The AQMEII project specifically focuses on regional modeling domains over Europe and North America (Galmarini et al., 2017), within which aerosol meteorology interactions was studied (Forkel et al., 2015; Makar et al., 2015a, 2015b; San Jose et al., 2015) over Europe and North America."

Line 106: 'Various multi-scale models: Are there more than the five model and seven simulations described here?

Response: Yes. This sentence briefly describes a bigger picture of MICS-ASIA Phase III, not

just Topic 3. Other topics include other models, such as GEOS-Chem. To avoid misunderstanding, I add one more sentence in the context: "A detailed overview of MICS-Asia Phase III, including descriptions of different research topics and participating models, will be published in a companion paper".

Lines 114 –119: The references should not be restricted to Chinese authors.

Response: We include more references on this topic: "Forkel et al., 2015; Makar et al., 2015a, 2015b; San Jose et al., 2015" from AQMEII Phase 2.

Line 125: The paper claims to be 'serving as the main repository of the information linked to Topic 3 simulations and comparisons.' To achieve this aim, more details must be added to the descriptions of those models where die description is quite. Please add also some paragraphs in section 2.1 (similar to section 2.5) in order to make this section better readable.

Response: Sorry for the relatively poor presentation in section 2.1. We have reorganized section 2.1 following the format in section 2.5 to make it more readable.

Table 1: The numbers attached to 'WRF-Chem' and 'NU-WRF' are unnecessary and confusing. Therefore, they must be removed. On the other hand, the model version is an important information which must be added wherever applicable. The contents of the table are not precise: For example, for M1 'RACM' must be replaced by 'RACM-ESRL' and 'MADE' by 'MADE/VBS'. What climatological data are applied as boundary conditions for M6? Please consider to add also information about the details of the radiation calculation. As shown by Curci et al. (2015, Atmospheric Environment, Vol. 115) the inherent assumption have a strong influence on the calculated AODs.

Response: Thanks for these great suggestions. We delete the numbers attached to WRF-Chem and NU-WRF and add the model version. We replaced 'RACM' and 'MADE', and add climatological boundary data for each model. We add both longwave and shortwave parameterization information and mixing state information for each model to Table 1 and cite Curci et al. (2015) to emphasize the importance of radiation calculation on AOD.

Table S1: Please check also whether this table needs to be more specific (similar to

Table 1). Please add also information about the two models which are missing in the table. Please consider also moving this table into the main part of the paper.

Response: To make it more specific, we add information for other missing models, including

climatological boundary data. We also merge S1 and Table 1.

Line 338: Please mention the source of the climatological data here or in Table 1.

Response: We add this in Table 1 in the revised manuscript.

Line 281: Please add the information which model uses the prescribed BVOC emissions and which one does the internal calculation (could also be added to Table 1).

Response: We add this in Table 1 in the revised manuscript.

Section 2.1: Please mention clearly in the text which aerosol meteorology interactions are switched on for each model.

Response: We include this information in Table 1 and add this in the revised manuscript.

Section 2.2: Please add some information on soil dust emissions (also to be included in Table 1).

Response: We add this information in Table 1 in the revised manuscript.

Section 2.3: Please include some information about the meteorological boundary conditions.

Response: We add this information in Table 1 in the revised manuscript.

Section 2.3: Please consider to add some information about the differences between the boundary values from the different data sources for selected variables (eventually to be included in the supplementary material). Looking into the boundary values may also help to understand differences between results of the different model runs.

Response: We add this information in the supplementary material as shown below. For aerosol species and gases except ozone, boundary conditions from the two models are pretty similar to each other. MOZART simulated ozone is higher than GEOS-Chem. Previous tests have showed that the influences of different boundary conditions have negligible impact on PM simulations, but larger impact on ozone (Abdalah et al., 2016). Topic 3 focuses on aerosol-weather-climate interactions in North China, so the impacts of different chemical boundary conditions are not quite important on our results. But it might be part of reason for poor ozone performance for some models. Thanks for this great suggestion. We have added this discussion in the revised paper.

Abdallah, C., Sartelet, K. and Afif, C., 2016. Influence of boundary conditions and anthropogenic emission inventories on simulated O 3 and PM 2.5 concentrations over Lebanon. Atmospheric Pollution Research, 7(6), pp.971-979.

[Figure]

Figure Monthly mean near surface CO, ozone, sulfate, BC and OC from GEOS-Chem (a-e) and MOZART (f-i)

Line 389: This sentence is confusing. According to Table S1 all simulations considered here are performed with aerosol meteorology interactions switched on. Were the simulations additionally performed also without aerosol meteorology interactions for the investigation of feedback effects? Anyway, this could be mentioned in the introduction as well as in the outlook, but not at this place.

Response: Yes, Table S1 only specifies what kind of interactions were turned on. We also performed without interactions to check the differences. The differences between with and without interactions will be present in paper part II on aerosol-meteorology interactions. Following your suggestion, we move this sentence to the introduction section.

Line 392 –393: Why are abbreviations given for temperature, humidity, and wind, and units for the shortwave radiation?

Response: To make it easier to show in figures.

Section 3: Why is the year 2013 described here, if it not discussed in the rest of the paper? Figure 3 could eventually be moved to the supplementary material.

Response: Following your suggestion, we move it to the supplementary material and remove corresponding sentences in the text.

Section 4.1: Cloud optical depth and integrated liquid water are important and should also be discussed (even, if no observational data are available).

Response: Following your suggestion, we add plots of integrated liquid water. We found this plot is good to explain why SWDOWN in north China is extremely low for M6. We added this point in the revised manuscript.

[Figure]

Line 427: This topic is not addressed in this paper.

Response: Thanks for pointing out. We delete it here and will provide discuss it the forthcoming companion paper.

Line 473: What is the reason for the bad performance of M6 and M7? Please check also cloud cover. And why is this huge difference in radiation between M6 and M7 not reflected in T2? Why does M7, which is also WRF, show such a large difference to M1 and M2?

Response: The reason for M7 overestimation of SWDOWN in North China is the exclusion of aerosol-radiation interactions, which tends to overpredict SWDOWN especially in high PM days and areas. T2 looks fine because T2 nudging was used, so T2 is close to measurements in M7. M1 and M2 used very different settings, with aerosol-radiation interactions, and without using meteorology nudging. M6 simulates a lower SWDOWN due to overestimation of cloud integrated liquid water as mentioned above. Sorry for the confusion here. We add the descriptions of setting in Table 1 and text to make it clearer.

Line 480: Why is this the case? (Figure 5e), M6 and M7 show a better consistence with observations than over northern China sites.

Response: We add plots of integrated liquid water, which may explain a better simulation of

M6 in southern China, and M7 shows a better agreement with observation because relatively lower PM level and weaker aerosol radiative effect in southern China.. We added this point in the revised manuscript.

[Figure]

Section 4.2: Spatial distributions of the gas phase pollutants (similar to figures 8 and 9) could be shown also in the supplement.

Response: Thanks for this suggestion. We have added in the revised manuscript.

[Figure]

Line 498: The 'only' is probably placed wrong here?

Response: Yes, we have removed it.

Line 504: Please give some evidence for this.

Response: This statement is based on the mean MBE averaged over all used CARE-China sites. It is typo here. We changed Beijing to "CARE-China sites".

Line 521: Please give more evidence for this statement. And what is the contribution of soil dust to PM10? What is the contribution of different of the dust emission parameterizations?

Response: In winter of the north Huabei Plain, soil dust generally contributes about 10% to PM concentration, but there is also primary PM from anthropogenic activity, such as power plant, traffic, construction etc. and this part of PM mostly settles in coarse mode, which may contribute to $PM_{10}$, but it's not clear if all the models include this emission sector and this sector is of higher uncertainty compared with other anthropogenic emission sectors. The schemes of dust deflation are similar in WRF-Chem series and different in M5, M6 and M7, the discussion on different dust schemes is less important given the relatively small contribution to $PM_{2.5}$. However, we added the dust implementation in Table 1 to provide how it affect PM10, which are shown below. The implementation of wind-blown dust are mostly significant in northwestern regions, and less important in Beijing and surrounding regions.

[Figure]

Line 524 – 526: What is the reason for the negative correlation? On an hourly basis, the diurnal course of ozone should reflect the course of the solar radiation. Or are the correlations just calculated from on daily values? Please clarify. How do the diurnal courses look like for the

individual models? Could the parameterization of dry deposition or differences in the lateral boundary conditions or differences in biogenic VOC emissions explain the overestimation of simulated ozone for M3 and M4? It is surprising, that the overestimated ozone seems not to be related to solar radiation, so there are probably other reasons for this overestimation. How do the ozone profiles look like, can they contribute to an explanation?

Response: We think the major causes for O3 overprediction in M3 and M4 are the combined effects of vertical diffusion and lateral boundary condition. M3 and M4 could predict larger vertical diffusivity coefficients, which leads to stronger vertical mixing, this not only results in lower NOx concentration and weaker titration of O3, but also stronger mixing of O3 from lateral boundary down to near surface, consequently causing higher O3 concentration than others.

Line 530: If the citation does not fit, it should be removed. Knote et al. (2015, Atmospheric Environment, Vol. 115) may probably suit better as this paper includes also a winter episode.

Response: Thanks. We replaced this citation with Knote et al. 2015.

Sections 4.3 and 4.4: It would be nice if soils dust were also included in the discussion.

Response: Thanks. We have added discussion of dust in the revised manuscript.

Lines 550 – 554: These general remarks are not necessary here.

Response: We remove this part.

Line 561: 'M5 and M6 shows : : :': Please observe the proper use of singular and plural (not only here but throughout the paper, e.g. line 582).

Response: We have changed these two and checked through the manuscript.

Lines 562 – 578: It is not clear in how far these statements apply to the models which are discussed here.

Response: We think these sentences are important to explain the possible reasons for the consistent underestimation of sulfate concentration from most of the models. Some important chemical processes for sulfate formation during hazy events are not implemented or fully considered in the participating models. M5 and M6 don't show the low biases possibly due to other compensatory processes, such as larger chemical reaction rate or lower deposition in the models.

Line 568 – 572: This sentence is quite hard to understand. Please split it into two or more sentences. Please also split other lengthy convoluted sentences.

Response: We have changed these two and checked through the manuscript.

Line 593: Is this just a general remark or is there some evidence for this? Does M7 really include heterogeneous nitrate formation?

Response: Yes, in default CMAQ, the heterogeneous nitrate related reactions include $N_2O_5$ +$H_2O$ and $2NO_2 + H_2O$

Line 608 and line 772: How can this be? Emissions were supposed to be the same for all models.

Response: We explain this in response to the same question from another reviewer. Emission should be similar among models, but treatments of deposition and aging process are different, we rewrite the sentence.

Line 609: Was the vertical distribution of the emissions not prescribed? If there are differences in the vertical distributions, they must be described.

Response: All the models release emission at the surface. We have rewritten this sentence.

Line 616: According to line 191, M2 does not include SOA formation

Response: We have removed M2 here.

Line 616 – 617: The statement about SORGAM does not fit here as M1 includes a

VBS approach (Ahmadov, R., et al., 2012, J. Geophys. Res.).

Response: Sorry, we double check with the modeler, and they confirm that SORGAM instead of VBS is used. We have changed the description in Table 1 and in the text.

Line 619: 'volatile' seems to be missing here

Response: We changed it to semi-volatile VOCs

Lines 619 – 624: These lines include quite general statements. How are they related to the models discussed in this paper?

Response: We have deleted this sentence.

Lines 642 – 643: This should have been mentioned in section 2.2.

Response: We have added the dust parameterization in section 2.2.

Lines 669 – 672: Please avoid this kind of redundancies (not only here).

Response: We have removed it and checked through the paper.

Lines 677 – 680: What is the contribution of soil dust during these situations?

Line 688: Please split this sentence.

Response: We have splitted.

Line 690 – 691: This statement is true, but unnecessary.

Response: We have removed.

Lines 692 – 698: These are quite general statements, which must be related to the applied models.

Response: We have removed

Line 710: M2 does not have modes

Response: We added (size bins)

Line 712: What are the consequences of this? If necessary, this can be discussed based on the findings by Curci et al. (2015, Atmospheric Environment, Vol. 115).

Response: Thanks for this good suggestion. We added the conclusion from Curci et al. (2015) that external mixing can increase simulated AOD but core-shell assumption is a minor issue.

**Figures and figure captions:**

Figure captions in general: More detailed descriptions must be given in the figure captions (are daily values or hourly values shown, relevant area, etc., depending on what is shown in the figure).

Response: Thanks for pointing out. We have added these information.

Figure 1: It looks like the labels M3 and M4 in the legend of Figure 1 are mixed up

(According to Table 1 and Figures 8 and 9 M4 is the small domain). Furthermore, using the same colors for the model domains as for the curves in Fig. 5 etc. would make the reading a bit easier.

Response: Thanks for pointing out. We noticed it after submission. We have changed it.

Caption of Figure 1: Repeating the model names in the caption (for example 'M1:

WRF-Chem, 45 km; M2 WRF-Chem, 50 km, M3 : : :') would make the paper a bit 'reader-friendly'.

Response: Thanks for this good suggestion. We have added: M1: WRF-Chem 45km; M2: WRF-Chem 50km; M3: NU WRF 45km; M4: NU-WRF 15km; M5: RIEMS-IAP 60km; RegCCMS 50km; WRF-CMAQ 45km

Caption of Figure 5: Please mention that this is a spatial average (for a, b and c: over which area) of daily values and explain error bars.

Response: Thanks for this good suggestion. We have added: (spatial daily values are averaged over measurements shown in S4 and S5; the error bars show the standard deviation of values over the measurement sites)

Figure 7: Please show also CO and PM2.5 although no measurements are available.

Response: Thanks for this good suggestion. We have changed.

Figures 8 and 9: Please include also PM10.

Response: Thanks for this good suggestion. We have changed.

Figures 8 and 9: The split into two figures appears quite arbitrary. Perhaps it would look better if the figures are organized differently: One figure with 7 rows (M1 – M7) and 3 columns (PM10, PM2.5, BC) and one figure with 7 rows (M1 – M7) and 4 columns (SO24, NO3, NH+4, OC) – no obligation to do this, just a suggestion.

Response: Thanks for this good suggestion. We have added PM10 in Figure 9.

Spatial distributions of the gas phase pollutants (similar to Figures 8 and 9) would be nice in the supplement.

Response: Thanks for this good suggestion. We have added gas pollutants.

Figure S4: This figure seems to be contorted. Please improve the quality. Why are all the lines within the single 'height groups' (e.g. at '1 km') at different heights? Please explain in the figure caption. Also: explain what is shown here (daily values, hourly values, : : :?)

Response: Do you mean Figure S7? I used slight differences to better show there values, otherwise they overlap with each other. The shown values are monthly mean. I have added these information (near surface observation is at 55m and model predictions are at 2m; comparisons are conducted at near surface, 1km and 3km; shifts in heights are made to make it clearer to avoid overlapping) in the revised caption and replotted it to sole the contortion problem.

---

## Author Response (AR2)

Dear Editor:

We would like to thank the editor and two anonymous reviewers for their constructive comments, which significantly improved this manuscript. Below, we address all comments point-by-point in **blue**.

**Anonymous Referee #2.**

General:

The paper has improved significantly as compared to the first version. Nevertheless, there are still some issues where the paper needs further improvement before it can finally be published. Although the majority of the reviewer comments are sufficiently addressed there are still some aspects which must be highlighted in more detail. Another deficiency of the paper is still the language quality of some (but not all) parts of the paper. Here, the authors with better knowledge of the English language as well as the native speaker are requested put some effort into improving those sections which are still not well written or a language editing service should be consulted.

Response: We (including native speaker) have read the manuscript very carefully and edited the language. Hope the language quality has been improved.

Detailed comments:

Line 43: Differences among the simulated chemical composition of the aerosol are not solely due to different description of the chemistry.

Response: We removed ", which is due to different parameterizations of chemical reactions" and added "However, it was also found that the ensemble mean of the models produced the best prediction skill in most cases. While this has been shown for other conditions (for example prediction of high ozone events in the US, this is to our knowledge the first time it has been shown for heavy haze events" in the abstract.

Page 9 - 11: References are not complete please check (not only for these pages but everywhere). For example, references are missing for RADM2, RACM, MADE/SORGAM, AE6.

Response: We add the references for RACM (Stockwell et al., 1997), RADM2 (Stockwell et al., 1990),MADE/SORGAM (Ackermann et al., 1998, Schell et al., 2001), AE6 (Carlton et al., 2010).

[revised manuscript text omitted]

Line 201: This sentence sounds like this was done recently, which is not the case.

Response: We change the sentence to "coupled by Schell et al (2001) was used in M1"
Schell, B., Ackermann, I.J., Hass, H., Binkowski, F.S. and Ebel, A., 2001. Modeling the formation of secondary organic aerosol within a comprehensive air quality model system. Journal of Geophysical Research: Atmospheres, 106(D22), pp.28275-28293.

Line 257 – 258: And what about RADM2 and RACM-ESRL?
Response: We add one sentence in the manuscript to describe this: "Speciation mapping of NMVOC emissions for groups using other gas-phase chemical mechanisms, such as CBMZ, RADM2 and CBM4, used the speciation framework documented in Li et al. (2014)."

Li, M., Zhang, Q., Streets, D.G., He, K.B., Cheng, Y.F., Emmons, L.K., Huo, H., Kang, S.C., Lu, Z., Shao, M. and Su, H., 2014. Mapping Asian anthropogenic emissions of non-methane volatile organic compounds to multiple chemical mechanisms. Atmospheric Chemistry and Physics, 14(11), p.5617.

Section 2.2: As some of the models account for dust, a short paragraph on dust emissions should be added as well.

Response: We add the following subsection to describe dust emissions.

2.2.6 Dust emissions

In M2, the Air Force Weather Agency (AFWA) version of the GOCART dust model was used. It calculates the saltation flux as a function of friction velocity ($u_*$) and threshold friction velocity ($u_{*t}$):

$$Q = C \frac{\rho_0}{g} u_*^3 \left(1 + \frac{u_{*t}}{u_*}\right)\left(1 - \frac{u_{*t}^2}{u_*^2}\right) \text{ when } u_* \geq u_{*t}$$

where $C$ is a tunable empirical constant, $\rho_0$ is air density, and $g$ is gravitational acceleration. The bulk vertical dust flux is estimated by $F = \alpha Q E$ (Marticorena and Bergametti, 1995), in which $\alpha$ is the sandblasting efficiency and $E$ is the dust erodibility factor. The erodibility factor data is included in the model geography dataset. In M3 and M4, the dust emissions are estimated using the GOCART dust model (Ginoux et al., 2001), which was determined by soil texture, moisture and surface wind speed. The drier the soil and the stronger the wind, the higher dust emissions over the regions where the erodibility factor is not zero. In M5, soil dust emissions were estimated by the approach from Han et al. (2004):

$$F = C_0 u_*^4 \left(1 - \frac{u_{*t}}{u_*}\right)(1 - f_i R_i) \text{when} u_* \geq u_{*t}, \text{ RH} \leq \text{RHt}$$

$C_0$ is a constant (1.4×10-15), $R_i$ is the reduction factor and $f_i$ is the factional coverage of i type of vegetation in a model grid (considering that vegetation cover can reduce dust emissions). u* and u*t are the friction and threshold friction velocities. RH and RHt are the relative humidity and threshold relative humidity near the surface. The total dust emission flux is apportioned to each size bin based on field measurements of vertical dust flux size distribution s in Chinese deserts.

Line 377: Why was this averaging done?

Response: Observations from a large number of stations are being used for meteorological comparisons. It is hard to show site-by-site comparison, so we use averaged and stand deviation to show spatial variations. Site-by-site comparisons were done in our previous papers focusing on the North China Plain, and the results show similar conclusions as this study: temperature and water vapor are simulated well, but wind speeds are overestimated during haze periods (Gao et al., 2016a).

Gao, M., et al. (2016a). "Modeling study of the 2010 regional haze event in the North China Plain." Atmospheric Chemistry and Physics16(3): 1673-1691.

Lines 462-463: Are the high values of the liquid water path simulated by M6 over Northern China really realistic? It looks like almost cloudless conditions are simulated by all the other models. What is the reason for these extremely high values of the LWP simulated by M6?

Response: They might not be realistic. The cumulus cloud physics and microphysics used by M6 are outdated and different from other models (Grell 3D was used by all other models), which might be the reason for different predictions of liquid water path.

Line 463: Please explain the reason for the poor performance of M7?

Response: The results shown for M7 is from simulation that aerosol-radiation was not turned on turned on (offline), so M7 radiation are higher than other models. We add this information in the revised manuscript "Online coupled WRF-CMAQ only considers aerosol-radiation interactions but no aerosol indirect effects. The WRF-CMAQ results shown in this paper are from an offline simulation (aerosol-radiation interaction was turned off)."

"The slightly higher daily maximum SWDOWN from M7 than other models is due to the deactivation of aerosol-radiation interactions in the presented M7 simulation."

The online results of WRF-CMAQ will be presented in the companion paper part II.

Lines 518-544 and response to reviewer #2: Not all reviewer comments are addressed in the revised version and in the response. Still no proper explanation is given for high ozone concentrations simulated by M4 and M4. Also, it is not discussed whether this difference is related to the maximum values or whether the concentrations during nighttime are not well reproduced. Neither the revised text nor the response to the reviewer gives an answer to this question. The attempt of an explanation, i.e. larger vertical diffusion, which results in less titration of ozone by NO, cannot really explain the higher ozone concentrations since the NOx concentrations simulated by M3 and M4 are quite similar to the concentrations found for M1. The authors might carefully check whether the implementation of RADM2 into NU-WRF is identical to the implementation of RADM2 with chem opt=1 into WRF-Chem. If this is the case, then titration of ozone by NOx is underestimated for regions with high NOx (see Forkel et al., 2015, see Figure S1 and the Appendix of that paper). However, this explanation must not be adopted uncritically by the authors – it is only valid if NU-WRF uses the same solver for RADM2 as it is used for chem_opt=1 in WRF-Chem!

Response: Thanks for these good suggestions. We checked the results that the difference is related to the too high concentrations during nighttime. We have confirmed with the NASA modeling group that the implementation of RADM2 in NU-WRF is identical to it in WRF-Chem. They just used the same codes in WRF-Chem v3.5.1 package: module_data_radm2.F and module_radm.F. In East China, where NOx concentrations are even higher than in Europe (Forkel et al., 2015), so this is the main reason for overestimation of ozone in M3 and M4. We have added these explanations in the revised manuscript: "The overestimations of ozone concentrations from M3 and M4 primarily occur during nighttime, implying the underestimated titration of ozone by NOx. Forkel et al. (2015) reported that the RADM2 solver in WRF-Chem has the problem of underestimating ozone titration in areas with high NO emissions, which is the version that applied in M3 and M4."

Line 647: The reference Zhao et al. (2015) is missing
Response: We add the following one:
Zhao, B., Wang, S., Donahue, N.M., Jathar, S.H., Huang, X., Wu, W., Hao, J. and Robinson, A.L., 2016. Quantifying the effect of organic aerosol aging and intermediate-volatility emissions on regional-scale aerosol pollution in China. Scientific reports, 6, p.28815.

Line 670: Compositions cannot be high. Please reword
Response: Thanks for pointing out. We add "mean concentrations of" before "PM$_{2.5}$ chemical compositions".

Lines 693 – 698: Are these small differences in the correlation coefficient really relevant?
Response: Thanks for pointing out. We delete these sentences of correlation coefficient.

Lines 717 – 725: All the mentioned concentrations are near surface values whereas AOD reflects vertically integrated values. Please add some comment about this issue.
Response: Thanks for this great suggestion. We change the sentence to "which can partially explain the largest simulated AOD by M2" and add another comment "The largest simulated AOD by M2 could also be related to different vertical distributions of aerosols".

Lines 782 – 783: This is just a rather general statement. The role of deposition is not discussed in more detail earlier in the paper.
Response: We delete this sentence in the revised manuscript.

Table 1: 1) Does 'Not available' mean 'not considered in the simulation' or just 'not supplied'? 2) What is the meaning of the blank spaces in the table (microphysics and surface physics for WRF-CMAQ)?
Response: NA means not considered in the simulation here. I add this description under the Table. I fill in the blank spaces in the revised manuscript.

Caption of Figure 1: 'M6' and 'M7' is missing before RegCCMs and WRF-CMAQ, respectively.
Response: We add 'M6' and 'M7'.

Indicating the locations of those stations where results are shown in the paper or in the supplement in Figures 6 and 7 would be helpful.
Response: We have added names of those stations in Figure S6 and S7.

[Figure]

Figure S9 (and related text): The absence of aerosol cloud interactions does not necessarily mean that no cloud water is simulated. So, why is the liquid water path not shown for M7?

Response: M7 did not submit liquid water path results, but it would be similar to M1 and M2.

Caption of Figure S10: According to the text of the paper, the curves show M2.

Response: Thanks for pointing out. We have changed it to M2.

Response to reviewer #2: The explanations about wind-blown dust should also be included in the paper.

Response: We add descriptions of dust emissions in subsection 2.2.6 and include the responses to reviewer #2 in the paper: "In winter of the North China Plain, soil dust generally contributes about 10% to PM2.5 concentrations (He et al., 2014), but there is also primary PM from anthropogenic activity, such as power plant, traffic, and construction etc. The primary particles are mostly in coarse mode, which might contribute to PM10 concentrations, but are highly uncertain compared with other anthropogenic emission sectors".

In addition, we add the following discussions about wind-blown dessert dust: "The spatial distributions of predicted wind-blown dust from M5 are slightly different from M2 and M3, with lower concentrations over the Gobi desert (in west Inner Mongolia) (PM10 in Figure 8). M2 and M3 used similar GOCART dust emission schemes based on wind speeds and erodible areas, while M5 furtherly considered the dust reduction by vegetation cover, which could partially explain the relatively lower wind-blown dust predictions from M5".

Final question: What is the status of the companion papers? Please add some more information (first author, journal).

Response: We are still analyzing the model outputs for the companion papers. The authors would be the similar to those in this manuscript, and would be published in the same special issue in ACP. We will further discuss about the model results and paper writing in a workshop to be held in next month and the companion paper will be submitted soon in one or two months.

[revised manuscript text omitted]